# Tropical Attention:
# Neural Algorithmic Reasoning for Combinatorial Algorithms

**Baran Hashemi**
Origins Data Science Lab
Technical University of Munich
Munich, Germany
`baran.hashemi@tum.de`

**Kurt Pasque**
Naval Postgraduate School
Monterey, California
`kurt.pasque@nps.edu`

**Chris Teska**
Naval Postgraduate School
Monterey, California
`christopher.teska@nps.edu`

**Ruriko Yoshida**
Naval Postgraduate School
Monterey, California
`ryoshida@nps.edu`

## Abstract

*Can algebraic geometry enhance the sharpness, robustness, and interpretability of modern neural reasoning models by equipping them with a mathematically grounded inductive bias?* To answer this, we introduce Tropical Attention, an attention mechanism grounded in tropical geometry that lifts the attention kernel into tropical projective space, where reasoning is piecewise-linear and 1-Lipschitz, thus preserving the polyhedral decision structure inherent to combinatorial reasoning. We prove that Multi-Head Tropical Attention (MHTA) stacks universally approximate tropical circuits and realize tropical transitive closure through composition, achieving polynomial resource bounds without invoking recurrent mechanisms. These guarantees explain why the induced polyhedral decision boundaries remain sharp and scale-invariant, rather than smoothed by Softmax. Empirically, we show that Tropical Attention delivers stronger out-of-distribution generalization in both length and value, with high robustness against perturbative noise, and substantially faster inference with fewer parameters compared to Softmax-based and recurrent attention baselines, respectively. For the first time, we push the domain of neural algorithmic reasoning beyond **PTIME** problems to **NP-hard/complete** problems, paving the way toward sharper and more expressive Large Reasoning Models (LRMs) capable of tackling complex combinatorial challenges in Phylogenetics, Cryptography, Particle Physics, and Mathematical Discovery. The code is available at https://github.com/Baran-phys/Tropical-Attention/.

## 1 Introduction

The *tropical semiring* $\mathbb{T} := (\mathbb{R} \cup \{-\infty\}, \max, +)$ (or its "min-plus" variant) replaces ordinary addition by maximum and multiplication by addition [1]. Polynomials over this semiring evaluate to *piecewise-linear*, polyhedral functions. These are the main objects of study in *tropical geometry* that translates algebraic geometry into combinatorics, turning varieties into polyhedral complexes, with wide applications across the intersection of matroid theory, combinatorial optimization, auction theory, enumerative geometry [1–5], and recently Machine Learning [6–13]. Because it analyses the entire polyhedral structure of solutions rather than a single Euclidean point, tropical geometry is a

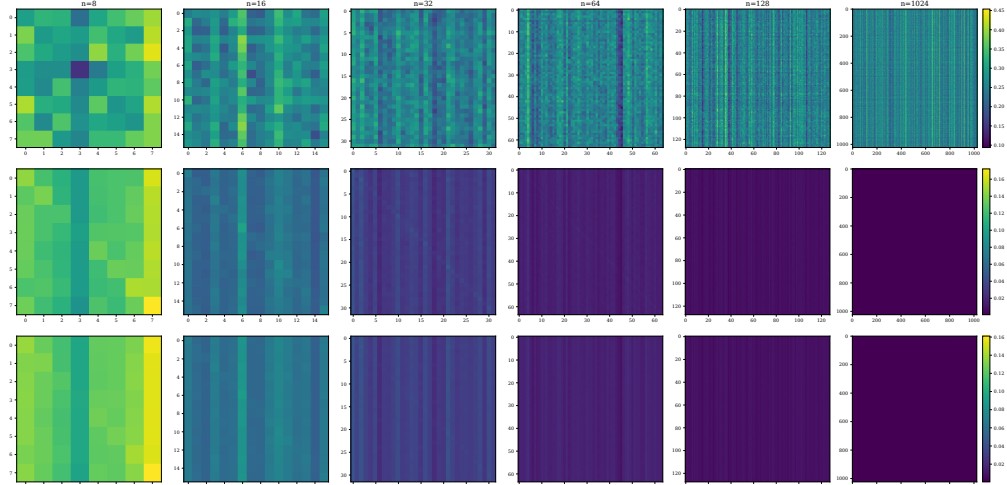

Figure 1: (top) **Tropical Attention** with sharp attention maps on learning the notorious [1] QUICKSELECT algorithm, showcasing a size-invariance and OOD lengths generalization behavior far beyond training ($8 \rightarrow 1024$). In contrast, both (middle) adaptive-softmax and (bottom) vanilla-softmax heads dilute and uniformly disperse as sequence length grows, failing to generalize. Each column evaluates the models on a new batch of independently drawn inputs of increasing length. Since the position of the target $k$-th element is different in each batch, the pattern of attention naturally changes to reflect the new data.

| Combinatorial Tasks | Baseline Transformers | | Universal Transformers (UT) | | Tropical Transformer |
| --- | --- | --- | --- | --- | --- |
| | *Vanilla* | *Adaptive* | *Vanilla w/ ACT* | *Adaptive w/ ACT* | |
| CONVEXHULL | $42.75^{\pm 2.06}$ | $48.25^{\pm 0.96}$ | 43.37 | 53.83 | $\mathbf{97.00}^{\pm 1.15}$ |
| KNAPSACK | $41.06^{\pm 1.76}$ | $39.18^{\pm 2.59}$ | 54.57 | 55.04 | $\mathbf{60.00}^{\pm 2.09}$ |
| QUICKSELECT | $4.66^{\pm 5.98}$ | $22.89^{\pm 2.49}$ | 37.05 | 40.44 | $\mathbf{77.06}^{\pm 3.78}$ |
| BINPACKING | $60.75^{\pm 2.49}$ | $64.25^{\pm 1.09}$ | 64.07 | 63.28 | $\mathbf{66.01}^{\pm 1.55}$ |
| SCC | $51.30^{\pm 3.91}$ | $56.50^{\pm 2.22}$ | 74.68 | 70.81 | $\mathbf{89.25}^{\pm 3.49}$ |
| SUBSETSUM | $21.13^{\pm 2.45}$ | $22.75^{\pm 5.25}$ | 41.43 | 42.05 | $\mathbf{87.50}^{\pm 6.45}$ |
| BALANCEDPARTITION | $80.55^{\pm 2.91}$ | $91.90^{\pm 5.52}$ | 80.01 | 91.13 | $\mathbf{96.73}^{\pm 3.50}$ |
| 3SUM | $80.00^{\pm 0.82}$ | $79.75^{\pm 0.50}$ | 81.12 | 81.67 | $\mathbf{82.75}^{\pm 1.59}$ |
| MINCOINCHANGE | $9.25^{\pm 1.86}$ | $17.98^{\pm 2.29}$ | 17.33 | 23.67 | $\mathbf{42.52}^{\pm 1.47}$ |
| FLOYD–WARSHALL | $12.81^{\pm 4.03}$ | $1.31^{\pm 0.36}$ | 7.59 | 0.97 | $\mathbf{0.81}^{\pm 0.08}$ |
| FRACTIONALKNAPSACK | $0.88^{\pm 0.06}$ | $0.86^{\pm 0.08}$ | 0.83 | 0.85 | $\mathbf{0.66}^{\pm 0.10}$ |

Table 1: Out-of-distribution Micro-$F_1$ score (top) and MSE for regression tasks (bottom) under **Length OOD** test. The **Tropical Transformer** outperforms all baselines, across combinatorial tasks while delivering $3\times$-$9\times$ faster inference and using $\sim 20\%$ fewer parameters than Universal Transformer (UT) [17] baselines with iterative attention that approximate the closure.

natural mathematical language for algorithms that must reason over *families* of inputs, particularly those generating such polyhedral structures. Dynamic programming (DP) exemplifies this connection. It is a cornerstone for numerous combinatorial optimization problems. The structure of these problems allows DP value functions to be described as piecewise-linear functions, forming polyhedral structures. They are just recursively constructed circuits over tropical semirings. Each such circuits, namely **tropical circuits**, compute monomials as feasible solutions over the underlying semiring making the DP update step effectively linear within this algebraic framework [14, 15].

---

[1]The challenge by Michael Galkin [16]

Many combinatorial optimization problems are specified by a family of these optimal solutions. For example, shortest-path algorithms such as Floyd-Warshall explicitly manifest this as tropical matrix products and closures, and their feasible solutions trace the faces of a polyhedral complex in parameter space [18]. More broadly, if candidate solutions to combinatorial algorithms are monomials (linear segments) in tropical space, then the computation is just a circuit of tropical gates. Several combinatorial algorithms like shortest paths [19, 20], change making [21, 22], knapsack [23–25], are nothing but recursively constructed tropical circuits. So the **tropical circuit model can give us a bridge between combinatorial optimization and neural architectures to perform sharp, piecewise-linear, and polyhedral computations,** offering a representation around which we can align the attention mechanism at the core of Transformer reasoning.

However, vanilla Transformers [26] are misaligned to this objective. Softmax-normalized dot-product attention lives in Euclidean geometry and produces smooth, quadratic decision boundaries. This smoothness blurs the hard $\arg\max / \arg\min$ structure on which combinatorial algorithms rely. As input length grows, softmax distributions can disperse, resulting in an increasingly flat probability distribution, a.k.a, dispersion [27] or attention fading [28]. Moreover, the exponential sensitivity of softmax makes logits vulnerable to small $\ell_\infty$ perturbations, harming adversarial robustness. As a result, Transformers equipped with softmax attention fail to extrapolate beyond the training regime of input length or magnitude on combinatorial tasks. For non-algorithmic reasoning tasks, even though injecting positional information [29–31] can alleviate the length extrapolation issue, we believe that the core of the issue lies within the attention mechanism itself.

**Our Contribution:** As a result, we propose *Tropical Attention*, a novel attention mechanism that incorporates tropical algebraic geometry to perform reasoning and information routing in tropical projective space. In tropical projective space, the attention scores are governed by the tropical Hilbert projective metric; consequently, the induced map is piecewise-linear, idempotent in the aggregation, and non-expansive (1-Lipschitz). In particular, Tropical Attention preserves the polyhedral structure characteristic of combinatorial value functions while inheriting the projective invariance and shortest-path geometry captured by the Hilbert metric [5]. Our viewpoint is aligned with Neuroalgebraic Geometry [32], in which neural computations are modeled by algebraic and tropical structures acting on polyhedral complexes. The mechanism is drop-in for standard Transformer blocks where reasoning occurs in tropical projective space and then returns to Euclidean coordinates for subsequent layers, so training and architectures remain unchanged.

We show that multi-head Tropical Attention (MHTA) realizes the target class of dynamic programs and approximates tropical circuits with *tropical transitive closure*. In particular, our expressivity theorems construct MHTA stacks that simulate tropical circuits without super-polynomial blow-ups in network size or the requirement for recurrent reasoning [17]. In other words, MHTA has sufficient capacity to approximate closure by absorbing the feasible solutions into its tropical polyhedral stratification, thus avoiding recurrent hidden state refinement [33].

Empirically on combinatorial tasks, including **NP-hard/complete** problems, this alignment translates into stronger out-of-distribution generalization together with substantial inference-time and parameter efficiencies relative to recurrent-style baselines. This study, also for the first time extends the applicability of Neural Algorithmic Reasoning [34] beyond the polynomial-time solvable problems to NP-hard/complete combinatorial optimization problems, such as KNAPSACK, BIN PACKING, and BALANCED PARTITION problems.

## 2 Background and Related Work

### 2.1 Out-of-Distribution Generalization

An important measure of a supervised learning model's reasoning is its ability to generalize to inputs that differ fundamentally from those encountered during training. This is known as out-of-distribution (OOD) generalization. Following [35] notations formally, let $\mathcal{X}$ denote the feature space and $\mathcal{Y}$ be the label set. A model $h : \mathcal{X} \to \mathcal{Y}$ is learned from training examples drawn independently and identically distributed from a training distribution $D_{\text{tr}}$ over $\mathcal{X} \times \mathcal{Y}$. Given a distinct test distribution $D_{\text{te}}$ on the same space, we define the OOD risk as, $\mathcal{R}_{D_{\text{te}}}(h) := \mathbb{E}_{(x,y) \sim D_{\text{te}}}[\ell(h(x), y)]$, where $\ell$

is a loss function. Its empirical estimate on a finite sample $S$ drawn from $D_{\text{te}}$ (i.e., $S \sim D_{\text{te}}^{|S|}$) is, $\widehat{\mathcal{R}}_S(h) := \frac{1}{|S|} \sum_{(x,y) \in S} \ell(h(x), y)$.

We say that the model $h$ OOD-generalizes from $D_{\text{tr}}$ to $D_{\text{te}}$ if its OOD risk $\mathcal{R}_{D_{\text{te}}}(h)$ remains comparable to its in-distribution risk $\mathcal{R}_{D_{\text{tr}}}(h)$, indicating minimal performance degradation despite the distributional shift. In the context of neural algorithmic reasoning, three main types of deviation between $D_{\text{tr}}$ and $D_{\text{te}}$ are important in measuring a model's capabilities:

**1. Length Generalization** Both distributions draw their numerical entries from the same range but the test sequences are strictly longer, $D_{\text{te}}(\mathcal{X}) \subsetneq \left(\mathbb{R}^{>0}\right)^{n_{\max}}$ with $n_{\max} > n_{\text{tr}}$. Here, a good performance indicates that the network has learned a parallel or recursive scheme that scales with input size rather than memorizing a fixed shallow circuit.

**2. Value generalization** The two distributions share the same support with respect to sequence length but $\text{supp}\left(D_{\text{te}}(\mathcal{X})\right)$ contains magnitudes never encountered during training, i.e. $\text{supp}\left(D_{\text{te}}(\mathcal{X})\right) \setminus \text{supp}\left(D_{\text{tr}}(\mathcal{X})\right) \neq \varnothing$ . For arithmetic or DP-style tasks, value generalization is the clearest evidence that the model has learned the *rule* rather than the lookup table of seen inputs.

**3. Perturbative-noise generalization** Noisy data, whether arising from measurement error, adversarial attack, or any other source, often causes models to make mistakes and must be accounted for in model design. To test noise robustness, $D_{\text{te}}$ is obtained from $D_{\text{tr}}$ by an $\ell_p$-bounded, perturbation map $\mathcal{A} : \mathcal{X} \to \mathcal{X}$ such that $x_{\text{ptb}} = \mathcal{A}(x)$ with $\|x_{\text{ptb}} - x\|_p \leq \varepsilon$. Robust generalization demands that the risk remains low even under the worst allowed $\mathcal{A}$. This regime probes the stability and smoothness of the learned function of the architecture. The perturbative noise robustness of Neural Algorithmic Reasoning models is very important for many real-world systems, especially for cryptographic schemes [36].

Length, value, and perturbative noise generalization stress complementary facets of algorithmic competence [37, 38]. Thus, a model as a true reasoning circuit [39, 40] that excels simultaneously in all three regimes offers strong evidence of having internalized the underlying combinatorial procedure rather than a brittle statistical surrogate.

## 2.2 Softmax Self-Attention Mechanism

Given an input sequence $\mathbf{X} = \left[\mathbf{x}_1, \ldots, \mathbf{x}_N\right]^\top \in \mathbb{R}^{N \times d_x}$, let $\mathbf{Q} = \mathbf{X}\mathbf{W}_Q^\top, \mathbf{K} = \mathbf{X}\mathbf{W}_K^\top, \mathbf{V} = \mathbf{X}\mathbf{W}_V^\top$, where the parameter matrices satisfy $\mathbf{W}_Q, \mathbf{W}_K \in \mathbb{R}^{d \times d_x}$ and $\mathbf{W}_V \in \mathbb{R}^{d_v \times d_x}$. Denote by $\mathbf{q}_i^\top$ and $\mathbf{k}_j^\top$ the $i$-th and $j$-th rows of $\mathbf{Q}$ and $\mathbf{K}$, respectively, and $\tau > 0$ for a temperature parameter. Vanilla self-attention computes, for every token $i$,

$$\mathbf{h}_i = \sum_{j=1}^N \alpha_{ij} \mathbf{v}_j, \qquad \alpha_{ij} = \text{softmax}_\tau\left(\langle \mathbf{q}_i, \mathbf{k}_j \rangle\right) := \frac{\exp\left(\langle \mathbf{q}_i, \mathbf{k}_j \rangle / \tau\right)}{\sum_{t=1}^N \exp\left(\langle \mathbf{q}_i, \mathbf{k}_t \rangle / \tau\right)}, \quad i = 1, \ldots, N, \quad (1)$$

where the $\text{softmax}$ is applied independently to each row of the score matrix $\mathbf{Q}\mathbf{K}^\top$. The temperature $\tau$ modulates the sharpness of the resulting probability vector, as $\tau \to 0$ the weights approach a one-hot selection, whereas large $\tau$ yields an almost uniform mixture. Equation 1 measures similarity with the Euclidean inner product, which is spherically invariant, meaning that every coordinate contributes equally, regardless of its algorithmic significance. Despite it's success in many tasks [26, 41], its geometric and numerical properties are ill-suited to algorithmic reasoning [35, 42]. We summarize the main shortcomings.

**1. Inherent blurriness** The exponential map assigns a non-zero weight to *every* token; even at low temperatures the second-largest term remains strictly positive. As problem size grows, the gap between the top two logits often decreases (e.g. when costs are drawn from a common distribution), so the resulting distribution cannot converge to a one-hot vector. In practice this leads to *soft* rather than decisive selections, hampering tasks that require exact order statistics [27, 28]. Recent diagnostic suites show that large language models fail on simple tasks of finding minima and second-minima even within In Distribution (ID) length tests [43, 44]. The attention kernel's inability to sharpen with scale is a primary culprit.

**2. Sensitivity to small perturbations** Because $\text{softmax}(z) \propto e^z$, a perturbation of size $\delta$ in the largest logit changes the corresponding weight by a multiplicative factor $e^\delta$. An adversary who

can alter a single entry of $\mathbf{QK}^\top/\tau$ by $\mathcal{O}(\log N)$ may invert the ranking of two tokens, propagating an $\mathcal{O}(1)$ error to downstream activations [45, 46]. This $\ell_\infty$-fragility persists even after common stabilisers such as temperature scaling or normalization layers [45].

**3. Mismatch with polyhedral decision boundaries** In a combinatorial optimization the value function is a tropical polynomial—piecewise linear with faces aligned to coordinate hyperplanes [5, 47]. The quadratic forms generated by Euclidean dot products carve the domain into spherical caps [48] rather than polyhedral cones; reproducing a DP recurrence therefore demands exponentially many heads or layers unless the desired structure is injected by hand.

**4. Temperature–gradient dilemma** Driving the distribution toward a hard $\arg\max$ necessitates lowering the temperature parameter $\tau$. Yet as $\tau \to 0$ the Jacobian of the softmax grows like $\tau^{-1}$, causing gradient explosion/vanishing [49]. Careful schedule tuning or gradient clipping becomes mandatory [45], adding hyper-parameter overhead.

### 2.3 Neural Algorithmic Reasoning

The problem of bridging symbolic algorithms and differentiable models has become known as *Neural Algorithmic Reasoning* (NAR). Neural Algorithmic Reasoning involves developing neural models and learning procedures to facilitate the *internalization* of algorithms directly in models' weights. Starting from early work [34] that aimed to demonstrate the applicability of Graph Neural Networks (GNNs) to approximate classical algorithms [50], the community has subsequently developed and expanded further in different directions [42, 51–58]. Some notable applications are constructing [59] and enumerating [60] combinatorial structures of a particular type [59], and dynamic programming [40, 61]. A fundamental objective of NAR is to achieve robust out-of-distribution (OOD) generalization through *algorithmic alignment*. Typically, models are trained and validated on small sets/sequences/graphs and tested on larger sets/sequences/graphs. This is inspired by classical algorithms' *size-invariance*, where correctness of the solution is maintained irrespective of the input size. Our work pursues this objective from a fresh, tropical geometric angle and provides universality guarantees and expressivity for an attention core within the NAR framework.

Recent work exclusively tried to quantify NAR failures when test sequences are longer [37, 38] or numerically larger [35], but have not assessed noise robustness scenarios. Perturbative noise itself is also of importance since real-world deployments must withstand the worst-case inputs and noise. Robustness in this setting is a test for whether a model has internalized genuine algorithmic structure rather than superficial statistical cues. Hence, we introduce *perturbative-noise generalization* as a third pillar for NAR benchmarking and show that Tropical Transformer demonstrates systematic gains across all three axes.

Based on the past related works, one can establish a demand for an attention mechanism that (i) is expressive and respects the underlying geometric structure of combinatorial algorithms, (ii) mitigates softmax dispersion that hampers OOD generalization, and (iii) delivers OOD noise robustness benefits. Tropical Attention positions itself at this intersection, drawing on a decade of tropical geometric insights to advance neural algorithmic reasoning. **To best of our knowledge, this level of incorporating Tropical algebraic geometry (not just pre-post processing arithmetic [13, 40, 62]) to define a new mathematically grounded reasoning architecture has not been done before.**

## 3 Tropical Attention: *Reasoning in Tropical projective space*

Tropical Attention arises from *combinatorial algorithm alignment*, that is information exchange is governed by order statistics, namely maxima, minima, and interval widths rather than absolute magnitudes. These operations live naturally in the tropical semiring, whose idempotence, translation covariance, and projective scale invariance match the properties of tropical geometry and it's computational mirror, tropical circuits. By contrast, the dot-product–softmax kernel depends on absolute scale and temperature, yields smooth (quadratic) decision boundaries, and exhibits dispersion as sequence length grows, thereby blurring the sharp structure required for decisive reasoning.

Our goal is to replace the *dot–product*, Softmax-based kernel of vanilla self-attention with a reasoning core that (i) takes place in the tropical polyhedral complex (ii) preserves the piecewise-linear geometry of combinatorial problems, and (iii) inherits the *1–Lipschitz* robustness of tropical linear maps. To do so, we perform a **tropicalization** map of queries, keys, and values to piecewise-linear projective

spaces carved out by polyhedral constraints, compute attention weights with the *tropical Hilbert projective metric* (See Appendix B and [5] for details on tropical algebraic geometry.), aggregate by a tropical matrix–vector product, and finally map the result back to Euclidean space so that the rest of the original algorithm (e.g Transformer) modular stack is untouched. We present the framework relating robustness and piecewise-linearity of maps and show how our proposed scheme offers improvements on OOD generalization tasks.

Let $\mathbf{X} \in \mathbb{R}^{N \times d}$ be the token embedding of the input. We define the *tropicalization map* by going to an amoeba representation of the input with a learnable valuation map, $\Phi : \mathbb{R}^{N \times d} \rightarrow (\mathbb{TP}^{d-1})^N$

$$\Phi_\lambda(\mathbf{X})_i = \mathbf{U}_i - \max_{1 \leq r \leq d} \mathbf{U}_{ir} . \mathbf{1}_d, \quad \text{where} \quad \mathbf{U} = \log\big(\max(\mathbf{0}, \mathbf{X})\big) \in \mathbb{R}^{N \times d} \tag{2}$$

for each row $i \in 1, \ldots, N$. The constant shift enforces $\max_i \phi_\lambda(\mathbf{x})_i = \epsilon$, so the output of $\phi_\lambda$ always lies in the tropical simplex, $\Delta^{d-1} := \big\{ z \in \mathbb{R}^d \big| \max_i z_i = \epsilon \big\}$, where every vector is projectively equivalent to exactly one point in the tropical simplex. In other words, $\Phi$ is a section of the quotient $\mathbb{R}^d / \mathbb{R}\mathbf{1}$.

**Lemma 3.1.** *For every embedded coordinate $i \in [N]$, the function*

$$v_\lambda(x) := \big[\phi_\lambda(x)\big]_i = \begin{cases} \log(x) - \lambda, & x > 0, \\ -\infty, & x \leq 0, \end{cases}$$

*where $\phi_\lambda$ is a (projective) valuation map. Hence the shifted map $\widetilde{v}(x) = v_\lambda(x) + \lambda = \log(\max(0, x))$ is an Archimedean valuation in the classical sense, and $\Phi$ is a matrix-valued valuation modulo tropical scalars; its image lies in the tropical simplex.*

After mapping each input token to tropical projective space, $\mathbf{Z} = \phi_\lambda(\mathbf{X}) \in \mathbb{TP}^{N \times d-1}$, we compute attention independently across $H$ heads.

**Definition 3.1** (Multi-head Tropical Attention (MHTA)). *Let $d_k = d/H$ be a fixed head dimension. Then, for every head ($h \in [H]$ one can choose learnable matrices $\mathbf{W}_Q^{(h)}, \mathbf{W}_K^{(h)}, \mathbf{W}_V^{(h)} \in \mathbb{R}^{d_k \times d}$ and define the tropical linear projections [7] $\mathbf{Q}^{(h)} = \mathbf{Z} \odot \mathbf{W}_Q^{(h)\top}, \mathbf{K}^{(h)} = \mathbf{Z} \odot \mathbf{W}_K^{(h)\top}, \mathbf{V}^{(h)} = \mathbf{Z} \odot \mathbf{W}_V^{(h)\top}$ where $\odot$ denotes max–plus matrix multiplication, $(\mathbf{A} \odot \mathbf{B})_{ij} = \max_t\{\mathbf{A}_{it} + \mathbf{B}_{tj}\}$. Then, using $d_\mathbb{H}$ the tropical Hilbert projective metric, defined in B.4, we will have the tropical attention score*

$$\mathbf{S}_{ij}^{(h)} = -d_\mathbb{H}\big(\mathbf{q}_i^{(h)}, \mathbf{k}_j^{(h)}\big), \qquad i, j \in [N],$$

*that comes with Projective Invariance and Non-expansiveness condition (discussed in B.4). Thereafter, the head outputs are aggregated via tropical matrix–vector product,*

$$\mathbf{C}_i^{(h)} = \bigoplus_{j=1}^{N} \mathbf{S}_{ij}^{(h)} \odot \mathbf{v}_j^{(h)} = \max_j\{\mathbf{S}_{ij}^{(h)} + \mathbf{v}_j^{(h)}\}, \qquad i \in [N].$$

*The tropical context picks the value that best aligns projectively with the query. Then, the contexts per head, will be mapped to the Euclidean domain via a smooth inverse map (devaluation ) $\psi(z) = \exp(z)$, and concatenated back to the original dimension, $\mathbf{H} = \big[\psi(\mathbf{C}^{(1)}) \| \ldots \| \psi(\mathbf{C}^{(H)})\big] \in \mathbb{R}^{N \times d}$.*

**Why Tropical Attention?** Every operation inside MHTA is piecewise linear and aligned with tropical geometry (not just tropical arithmetic). Hence the entire network computes a tropical polygonal map whose cells are polyhedral cut out by hyperplanes. This is aligned with combinatorial algorithms, whose solutions correspond to a vertex of a polytope and every decision boundary is a facet. Training a transformer with Tropical attention therefore starts from a hypothesis space that already mirrors the solution structure of a combinatorial algorithms. That is what we call a ***polyhedral inductive bias***. By contrast, Euclidean softmax attention inserts an exponential map, blurring the sharp decisions on their input data. Classical transformers modulate the entropy–versus–sharpness trade-off through a temperature parameter in the softmax; MHTA sharpness is built in and temperature-free. Moreover, since every intermediate representation of MHTA lies in the projective simplex $\Delta^{d-1}$, going through $\arg\max$ is well-defined (no equal maxima except on a set of measure 0) and is stable by global scaling meaning that shifting the entire vector by $\lambda \in \mathbb{R}^d$ does not alter which index attains the maximum. In other words, only relative relations between inputs matter, thus Tropical attention is inherently robust against distribution shifts.

Furthermore, each MHTA head can function as a tropical gate in a **tropical circuit**. A tropical circuit is a finite acyclic digraph whose input vertices store either a variable or a non-negative real constant, while every internal vertex has in-degree two and outputs the *maximum* or the *sum* of its two predecessors. The circuit's size is the number of internal gates. Classical pure DP algorithms are recursive tropical circuits of this kind; consequently, lower bounds for tropical circuits translate directly into limits for such DP schemes. An MHTA head can also be interpreted as a single tropical gate. A single head implements the composite transformation $(u, v) \longmapsto \max_j \{S_{ij} + v_j\}$, where the score $S_{ij}$ itself is obtained through several applications of $\max$ and $+$ gates. The outer maximization provides the $\oplus$-gate, while the summand $v_j$ furnishes a $\odot$-gate acting on two variable inputs. Thus every head is a compact, differentiable wrapper around the two tropical primitives, and a full multi-head layer is simply a collection of such gates operating in parallel on a shared input tape, creating a **tropical transitive closure**. Training a multi-layer MHTA therefore amounts to discovering how these gates should be wired together, rather than coaxing a Euclidean softmax kernel to emulate max–plus algebra indirectly. As a result of developing MHTA, we prove that it is a universal approximator of max-plus dynamic programming for combinatorial optimization with closure (Theorem C.3, Corollary C.3.1, and Theorem 3.2).

**Theorem 3.2** (Simulation of max–plus dynamic programs)**.** *Let* $(S, E)$ *be a finite directed acyclic graph with* $|S| = N$ *vertices and edge weights* $\{w_{uv}\}_{(u,v) \in E} \subset \mathbb{T}$*. Fix a source vertex* $v_0 \in S$ *and consider the max–plus Bellman recursion*

$$d_v(t+1) = \bigoplus_{u: (u,v) \in E} \big( w_{uv} \odot d_u(t) \big), \qquad d_v(0) = \delta_{v,v_0}, \quad t \in \mathbb{N}.$$

Theorem 3.2, Theorem C.3, and Theorems C.4 and C.5 show *upper bounds*, i.e., sufficient conditions, of $T$ and $N$ such that a stack of $T$ MHTA layers and $N$ heads can approximate any horizon tropical circuit for a dynamic program.

**Remark 3.1.** *A shallow MHTA is sufficiently expressive to approximate the tropical transitive-closure map, and thereby avoids explicit recurrence, to encode a very rich polyhedral geometry of combinatorial algorithmic reasoning into attention and provide a non-recurrent (one-shot) solution. However, in the worst-case scenario, the non-recurrent representation requires head-width proportional to the number of active path monomials up to* $\mathcal{O}(n^2 2^k)$*. Depth-(T) stacks realize the same computation with polynomial resources by distributing the computation across layers.*

In other words, Tropical Attention has sufficient capacity to approximate closure by absorbing the feasible solutions into its tropical polyhedral stratification, thus avoiding explicit recurrence. **Consequently, a MHTA module learns the closure directly rather than implementing the recurrence step by step.** By contrast, Recurrent Transformers such as the Universal Transformer [17] incorporate a depth-wise recurrence to learn the hidden state representation per token. The recurrent function evolves in parallel across token positions while exchanging information through self-attention. In the next section, Section 4, we show that, Tropical Transformer provides a stronger out-of-distribution generalization than Universal Transformer with Dynamic Halting mechanism, while delivering substantially faster inference with much fewer parameters.

## 4 Experiments

We evaluate Tropical transformers on eleven combinatorial tasks (see E), several of which are NP-hard/complete problems, and the the Long Range Arena (LRA) benchmark [63], a standard for testing transformers on long-sequence tasks across text, image, and math domains. For each combinatorial task we measure three complementary forms of out-of-distribution (OOD) generalization: Length OOD (longer inputs), Value OOD (unseen magnitudes), and noise robustness (perturbed inputs). A procedure to compare between vanilla attention and Tropical Attention is described in Appendix D. For our experiments we custom generated both train and test datasets following procedures from the canonical algorithmic reasoning benchmark CLRS [43]. This decision was due largely to the absence of NP-hard and NP-complete problems in the CLRS benchmark, but also because our framework is designed for sequence and set-based data modalities and our OOD evaluation includes two extra new evaluation pipelines, adversarial perturbations and value generalization. All datasets, generation scripts, and OOD protocols are described in Appendix E and F.

For our experiment we consider three variants, (i) **Vanilla**: Standard transformer encoder with softmax dot-product attention. (ii) **Adaptive**: Transformer equipped with adaptive softmax attention

from [27]. (iii) **Tropical**, which every attention block is replaced by MHTA. For length OOD tests, we also compare with 32-step Universal Transformer (UT) with dynamic halting, as a recurrent attention model, under the vanilla softmax and adaptive-temperature softmax kernels. To ensure a fair comparison, all variants share identical backbone hyperparameters: depth, width, and number of heads. The only architectural difference is the attention kernel. Crucially, no model sees OOD examples during optimization. We follow a uniform procedure in which each model is trained from scratch under the same training regime with task specific fixed input sequence lengths and value ranges.

**Out-of-Distribution Protocols** In order to assess OOD generalization, we construct three stress tests: (i) **Length OOD** – inputs drawn from the same value range but with longer input sequence lengths. (ii) **Value OOD** – the input sequence lengths are fixed and the values are sampled from an increasingly large range (for example, if the models trained on inputs sampled from the range $[-5, 5]$ an out of distribution evaluation would be inputs sampled from the range $[-10, 10]$). (iii) **Perturbative noise OOD** – the input sequence lengths are fixed and the values are from the same input range, but a subset of the input values are perturbed randomly.

# 5    Results and Discussion

Section 2.2 elaborated why and how softmax self-attention - and its descendants - are incapable of generalizing to OOD inputs in combinatorial problems, and Section 3 discussed *why* Tropical Attention can generalize in the combinatorial regime. With our experimentation, we seek to show *if* and, if so, *how* Tropical Attention generalizes in this domain.

To answer *if* Tropical Attention generalizes, we report the numerical results of our experimentation in Tables 1 and 3. The Tropical attention architecture achieves superior OOD performance to both the Vanilla and Adaptive softmax attention. Notably, this out performance can be seen in both regression and classification combinatorial tasks and across OOD protocols, validating our theoretical results from Section 3. The Trop-

Table 2: Average inference time per sample across all tasks and parameter count.

| Model | CPU (ms) | GPU (ms) | Params. |
|---|---|---|---|
| Vanilla UT w/ ACT | 6.285 | 0.027 | 50,242 |
| Adaptive UT w/ ACT | 7.898 | 0.018 | 50,242 |
| Tropical Transformer | **1.949** | **0.003** | **40,961** |

ical architecture's ability to generalize well across OOD protocols and problem sets, especially the notorious Quickselect, suggests that instead of simply learning the specific data it is trained on, these purpose-built models learn the underlying polyhedral structure of the combinatorial algorithm. The results, from Tables 1 and 2, demonstrate two key findings. MHTA shows a strong performance, even when compared against an iterative attention class of transformers with a dynamic Adaptive Computation Time (ACT) mechanism where they can approximate an algorithmic closure, the Tropical Attention model still achieves better OOD performance across all algorithmic tasks. The more interesting results are that our model achieves these results while being on average **3×-9× faster at inference** and using **20% fewer parameters** than the Universal Transformer baselines. Evaluating on the Long Range Arena (LRA) benchmark as shown in Table 4, Tropical Transformer achieves highly competitive, State-of-the-art (SOTA) results, placing *second* overall in average accuracy across the benchmark's tasks. This performance shows that the benefits of Tropical Attention stand as a viable and powerful mechanism for general-purpose sequence modeling.

In order to understand *how* Tropical Attention outperforms, we explore the tropical Attention maps relative to vanilla and adaptive attention maps for both Quickselect and Knapsack. On the QUICKSELECT task the goal is to find the $k$-th smallest elements. For such tasks, maintaining focus means the ability to allocate a high attention score to the correct items, creating sharp spikes in the heatmap. Contrarily, losing focus means the attention uniformly disperses across elements, with no single item receiving a high score. Figure 1, which depicts attention maps as sequence length increases for all models, shows that the vanilla and adaptive models exhibit attention fading. In contrast, the Tropical Attention consistently shows bright, distinct bands, indicating it continues to allocate sharp attention even at larger length sequences. Figure 2, replicating visualizations from [27], depicts a normalized attentional head for the Quickselect task for a batch of 32 sets, over the 8 items with the largest keys by the $\ell_2$-norm. If the head operates correctly, it must allocate sharp attention to the position of $k$-th smallest element. Again, we see that the attention on both softmax models quickly dilute/disperse as sequence length grows OOD while the tropical Attention maintains focus.

Table 3: Out-of-distribution performance under **Value OOD** and **Perturbative Noise** tests. Top: Micro-$F_1$ for classification tasks; Bottom: MSE for regression tasks.

| Algorithmic Tasks | Value OOD | | | Perturbative Noise | | |
|---|---|---|---|---|---|---|
| | Vanilla | Adaptive | Tropical | Vanilla | Adaptive | Tropical |
| ConvexHull | $22.75^{\pm3.59}$ | $23.77^{\pm3.10}$ | $\mathbf{\underline{34.25}}^{\pm1.71}$ | $90.75^{\pm2.22}$ | $91.00^{\pm2.16}$ | $\mathbf{\underline{96.00}}^{\pm2.16}$ |
| Knapsack | $38.87^{\pm3.43}$ | $26.92^{\pm1.33}$ | $\mathbf{\underline{49.67}}^{\pm2.01}$ | $67.85^{\pm3.19}$ | $68.36^{\pm3.51}$ | $\mathbf{\underline{74.67}}^{\pm3.13}$ |
| Quickselect | $74.22^{\pm2.30}$ | $\mathbf{74.30}^{\pm1.99}$ | $71.10^{\pm3.11}$ | $33.87^{\pm7.11}$ | $34.82^{\pm4.79}$ | $\mathbf{\underline{57.22}}^{\pm5.01}$ |
| BinPacking | $67.26^{\pm3.70}$ | $74.23^{\pm1.51}$ | $\mathbf{\underline{78.54}}^{\pm1.89}$ | $55.38^{\pm5.10}$ | $60.64^{\pm3.92}$ | $\mathbf{\underline{61.19}}^{\pm4.33}$ |
| SCC | $78.51^{\pm3.08}$ | $\mathbf{81.38}^{\pm2.62}$ | $74.86^{\pm5.01}$ | $70.00^{\pm5.98}$ | $\mathbf{71.33}^{\pm1.96}$ | $69.86^{\pm4.17}$ |
| SubsetSum | $34.75^{\pm6.60}$ | $28.50^{\pm10.12}$ | $\mathbf{\underline{79.25}}^{\pm5.38}$ | $3.75^{\pm1.50}$ | $3.00^{\pm1.63}$ | $\mathbf{\underline{72.75}}^{\pm10.01}$ |
| BalancedPartition | $\mathbf{63.40}^{\pm4.29}$ | $56.57^{\pm1.18}$ | $55.76^{\pm5.63}$ | $51.06^{\pm2.66}$ | $57.06^{\pm1.08}$ | $\mathbf{\underline{57.29}}^{\pm1.33}$ |
| 3SUM | $26.00^{\pm3.16}$ | $\mathbf{26.25}^{\pm3.50}$ | $22.00^{\pm2.16}$ | $47.50^{\pm9.47}$ | $49.25^{\pm9.22}$ | $\mathbf{\underline{65.25}}^{\pm3.59}$ |
| MinCoinChange | $23.64^{\pm4.07}$ | $2.20^{\pm1.13}$ | $\mathbf{\underline{33.18}}^{\pm5.64}$ | $22.12^{\pm2.75}$ | $18.44^{\pm4.49}$ | $\mathbf{\underline{33.75}}^{\pm4.89}$ |
| Floyd–Warshall | $87.68^{\pm5.65}$ | $56.30^{\pm3.04}$ | $\mathbf{\underline{55.30}}^{\pm4.36}$ | $7.54^{\pm3.63}$ | $5.29^{\pm2.56}$ | $\mathbf{\underline{4.39}}^{\pm1.62}$ |
| FractionalKnapsack | $0.24^{\pm0.12}$ | $0.17^{\pm0.03}$ | $\mathbf{\underline{0.08}}^{\pm0.03}$ | $0.05^{\pm0.02}$ | $0.03^{\pm0.01}$ | $\mathbf{\underline{0.02}}^{\pm0.01}$ |

Similarly, Figure 3 depicts length OOD on the full attention head for the Knapsack problem, a classic dynamic program corresponding to tropical circuits. Each model begins sharp in distribution, but the Tropical Attention head maintains the same activation pattern across each input length, strongly suggesting it has learned and internalized the underlying combinatorial problem and polyhedral structure vice the specific training data indicated by the vanilla and adaptive fading attention patterns.

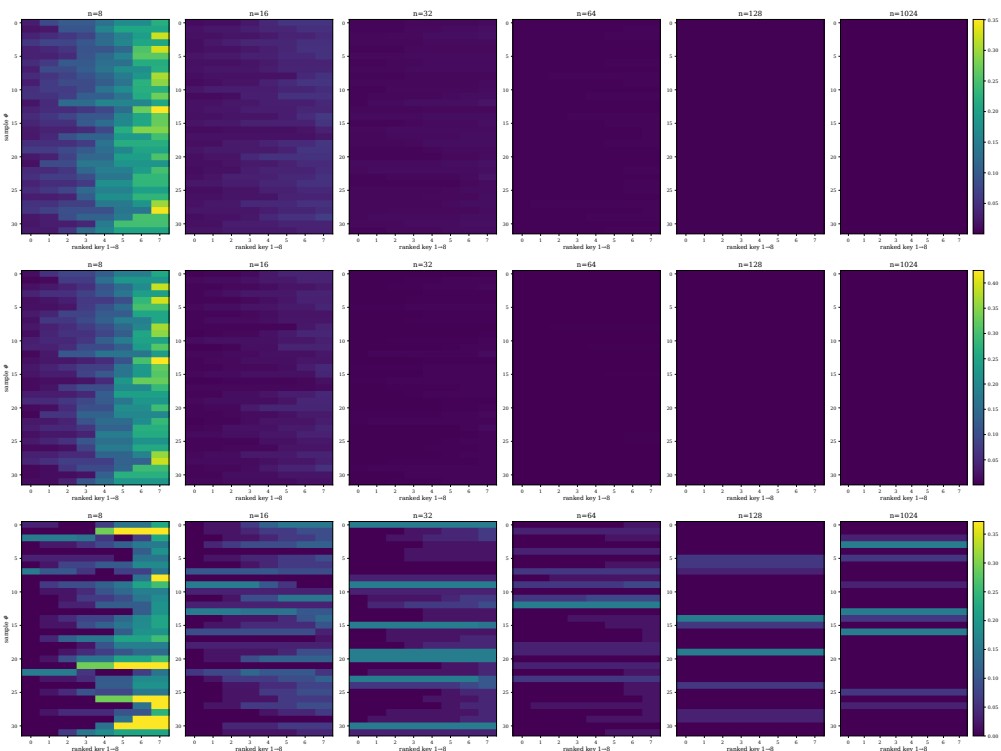

Figure 2: Stacked attention head representations for Quickselect under (a) Vanilla, (b) Adaptive, and (c) **Tropical** models. Each model was trained on length 8 sequences and was evaluated from Left to Right on length 16 to 1024 sequences. Each image was generated by a batch of 32 inputs. The columns are the 8 largest keys by $\ell_2$-norm. Heatmap values are the attention of the row item at the column key.

Table 4: We report classification accuracy for each task and the average accuracy across all tasks. All results are taken from their respective papers, except for Adaptive Softmax, which we re-implemented and evaluated.

| Models | ListOps | Text | Retrieval | Image | Pathfinder | Avg. | Complexity |
|---|---|---|---|---|---|---|---|
| Transformer [26] | 36.37 | 64.27 | 57.46 | 42.44 | 71.40 | 54.39 | $\mathcal{O}(n^2)$ |
| Longformer [64] | 35.63 | 62.85 | 56.89 | 42.22 | 69.71 | 53.46 | $\mathcal{O}(n)$ |
| Linformer [65] | 35.70 | 53.94 | 52.27 | 38.56 | 76.34 | 51.36 | $\mathcal{O}(n)$ |
| Performer [66] | 18.01 | 65.40 | 53.82 | 42.77 | 77.50 | 51.41 | $\mathcal{O}(n)$ |
| Elliptical [67] | 37.8 | 65.6 | 80.3 | 40.2 | 73.2 | 61.24 | $\mathcal{O}(n^2)$ |
| Fourierformer [68] | 40.73 | 75.02 | 85.35 | 53.17 | 83.43 | 67.54 | $\mathcal{O}(n \log n)$ |
| MEGA [69] | 63.14 | **90.43** | **91.25** | **90.44** | 96.01 | **86.25** | $\mathcal{O}(n \log n)$ |
| AdaptiveSoft. [27] | 47.15 | 75.52 | 79.56 | 51.58 | 80.94 | 66.95 | $\mathcal{O}(n^2)$ |
| Tropical Transformer | **68.65** | 70.13 | 64.82 | 60.04 | **97.33** | 72.79 | $\mathcal{O}(n^2)$ |

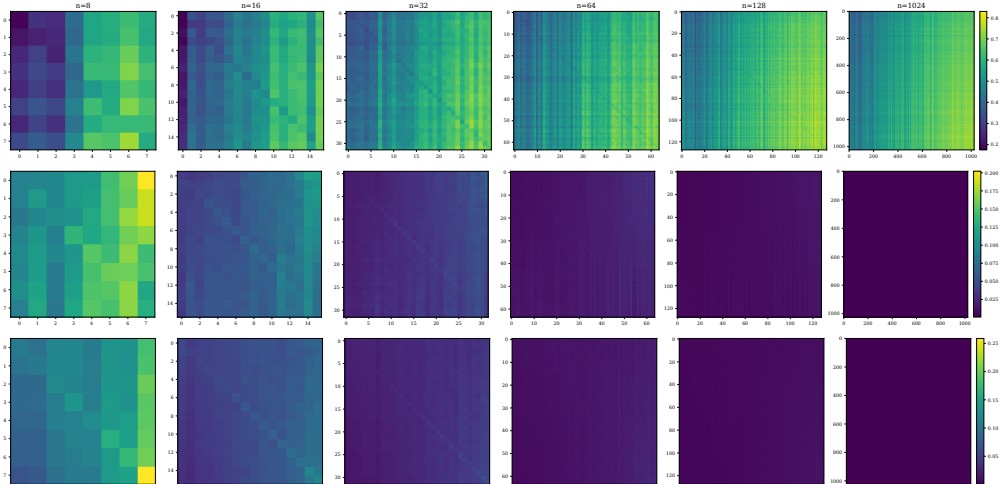

Figure 3: (top) **Tropical Attention** with sharp attention maps on learning the KNAPSACK algorithm, showcasing a size-invariance and OOD lengths generalization behavior far beyond training ($16 \rightarrow 1024$). In contrast, both (middle) adaptive-softmax and (bottom) vanilla-softmax heads dilute and disperse as sequence length grows, failing to generalize.

## 6 Conclusion

We introduced Tropical Attention, replacing softmax-normalized dot–product attention with an attention mechanism that operates in tropical projective space. On the theory side, we showed that multi-head Tropical Attention (MHTA) simulates tropical circuits and realizes tropical transitive closure via finite-depth compositions, with polynomial resource bounds (Theorem C.3, Corollary C.3.1, Theorem 3.2). These guarantees provide a principled account of scale-invariance and sharp, polyhedral decision boundaries, properties that are essential for reasoning models expected to generalize beyond their training distribution. Empirically, across various combinatorial problems, Tropical transformer achieved SOTA out-of-distribution generalization, and delivered stronger noise robustness, while being much faster at inference with fewer parameters than the recurrent/iterative attention baselines. These findings carry an important message for both neural algorithmic reasoning (NAR) and Large Reasoning Model (LRM) communities: **tropicalization of reasoning and going beyond softmax not only enriches the algorithmic power of attention mechanisms but also yields tangible gains on reasoning tasks.** We believe Tropical Attention opens compelling avenues for hybrid semiring architectures and for leveraging tropical geometry to reason over discrete structures within deep learning systems. Future work will explore sparse tropical kernels and applications to graph-theoretic domains, aiming for ever-stronger generalization guarantees in neural algorithm and reasoning synthesis.

## Acknowledgments and Disclosure of Funding

This research was supported by the Excellence Cluster ORIGINS, funded by the Deutsche Forschungsgemeinschaft (DFG, German Research Foundation) under Germany's Excellence Strategy – EXC-2094-390783311. B.H extends his gratitude to the organizers and the wonderful instructors, Marta Panizzut and Margarida Melo, of the 2024 Trieste Algebraic Geometry Summer School (TAGSS) on Tropical Geometry, where the idea of the project was sparked. K.P., C.T. and R.Y. are partially supported by NSF Division of Mathematical Sciences: Statistics Program DMS 2409819.

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

# A   Limitations

Although Tropical Attention is out performing in almost all algorithmic tasks, this study was conducted on combinatorial algorithms and we have not yet demonstrated how Tropical transformers can scale to perform on generative and autoregressive next-token prediction language tasks. In particular, the computational and memory overhead introduced by tropical operations and the tropical Hilbert metric could incur nontrivial runtime costs or scaling challenges.

# B   Tropical Geometry

The most fundamental component of tropical algebraic geometry is the *tropical semiring* $\mathbb{T} := (\mathbb{R} \cup \{-\infty\}, \oplus, \odot)$. The two operations $\oplus$ and $\odot$, called *tropical addition* and *tropical multiplication* respectively, are defined as follows.

**Definition B.1.** *For $x, y \in \mathbb{R}$, their* tropical sum *is $x \oplus y := \max\{x, y\}$; their* tropical product *is $x \odot y := x + y$; the* tropical quotient *of $x$ over $y$ is $x \oslash y := x - y$.*

For any $x \in \mathbb{R}$, we have $-\infty \oplus x = 0 \odot x = x$ and $-\infty \odot x = -\infty$. Thus $-\infty$ is the tropical additive identity and $0$ is the tropical multiplicative identity. Furthermore, these operations satisfy the usual laws of arithmetic, namely associativity, commutativity, and distributivity. The set $\mathbb{R} \cup \{-\infty\}$ is therefore a semiring under the operations $\oplus$ and $\odot$. While it is not a ring since it lacks an additive inverse, one may nonetheless generalize many algebraic objects over the tropical semiring, the study of these, in a nutshell, constitutes the subject of tropical algebra. In order to have a transition from classical arithmetic to tropical arithmetic we need a series of transition maps, which is referred to as *tropicalization*.

**Definition B.2.** *(The valuation map) Let $d \in \mathbb{N}$ and write $\mathbb{R}^d$ the field of real numbers. A valuation on $\mathbb{R}$ is a function* val: $\mathbb{R} \to \mathbb{R} \cup -\infty$ *satisfying the following three axioms:*

1. $\mathrm{val}(a) = -\infty \iff a = 0$;

2. $\mathrm{val}(ab) = \mathrm{val}(a) + \mathrm{val}(b)$;

3. $\mathrm{val}(a + b) \leq \max\{\mathrm{val}(a), \mathrm{val}(b)\} \; \forall \, a, b \in \mathbb{R}$.

One approach to tropical geometry, is to define a tropical variety as a shadow of an algebraic variety that involves logarithmic limit sets. Classically, the *amoeba* of a variety is its image under taking the coordinate-wise logarithm of the absolute value of any point on the variety [14]. The logarithm turns ordinary multiplication into tropical addition:

$$\mathrm{val}(x \odot y) = \mathrm{val}(x) + \mathrm{val}(y), \qquad x, y \in \mathbb{R}_{>0},$$

and satisfies the sub-additive inequality $\mathrm{val}(x + y) \leq \max\{\mathrm{val}(x), \mathrm{val}(y)\} + \log 2$. Hence val is a Archimedean log map up to a harmless additive constant. For an input $X \subset \mathbb{R}^d$ we call

$$\mathrm{Trop}(X) := \mathrm{val}(X) \subset \mathbb{T}^d$$

its *tropicalization*. All subsequent reasoning, including attention weight computations, will take place in this max-plus space. When $X$ is a smooth manifold, $\mathrm{Trop}(X)$ is typically a curved domain whose "tentacles" encode asymptotic directions of $X$. Passing to the max-plus algebra straightens those curves into polyhedral pieces, providing the piecewise-linear structure on which our Tropical Attention operates.

**Definition B.3.** *(The tropical projective space [1].) We regard $\mathbb{T}^d$ as a semimodule over the tropical semiring by coordinate-wise operations. Introduce*

$$\mathbf{1}_{d+1} := (1, \ldots, 1) \in \mathbb{R}^{d+1}, \qquad \mathbb{T}^{d+1} := (\mathbb{R} \cup \{-\infty\})^{d+1} \setminus \{-\infty\}^{d+1}.$$

*Declare two points $x, y \in \mathbb{T}^{d+1}$ projectively equivalent, written $x \sim y$, if there is a scalar $\lambda \in \mathbb{R}$ such that $y = x + \lambda \mathbf{1}_{d+1}$. The quotient*

$$\mathbb{TP}^d := \mathbb{T}^{d+1} / \sim$$

*is the* tropical projective space. *See [1] for more details on tropical geometry.*

Every class has a unique representative with maximal coordinate equal to 0, so $\mathbb{TP}^d$ identifies with the standard simplex $\Delta^d := \{w \in \mathbb{R}^{d+1} \mid \max_i w_i = 0\}$. Attention weights produced by the softmax surrogate live in the Euclidean simplex; Tropical Attention will instead output points of $\Delta^d$ interpreted tropically, guaranteeing sharp arg max behavior.

**Definition B.4.** *(The tropical Hilbert projective metric.) For $x := (x_1, \ldots, x_{d+1}), y := (y_1, \ldots, y_{d+1}) \in \mathbb{T}^{d+1}$ put*

$$d_{\mathbb{H}}(x, y) := \left(\max_i (x_i - y_i)\right) - \left(\min_i (x_i - y_i)\right) = \mathrm{diam}(x \oslash y),$$

*where $x \oslash y$ denotes the coordinate-wise tropical quotient $(x_1 - y_1, \ldots, x_{d+1} - y_{d+1})$ and $\mathrm{diam}$ its range.*

The metric descends to $\mathbb{TP}^d$ and enjoys two key properties:

1. **Projective invariance.** $d_{\mathbb{H}}(x + c\mathbf{1}_{d+1}, y + c\mathbf{1}_{d+1}) = d_{\mathbb{H}}(x, y)$ for all $c \in \mathbb{R}$.

2. **Non-expansiveness of max-plus-affine maps [70].** Every tropical linear map $A : \mathbb{T}^{d+1} \to \mathbb{T}^{m+1}$ is 1-Lipschitz: $d_{\mathbb{H}}(Ax, Ay) \leq d_{\mathbb{H}}(x, y)$.

These facts, due to Nussbaum and further developed by Akian–Gaubert, furnish tight robustness guarantees, perturbing the inputs by $\epsilon$ in Hilbert distance changes the output of any compositional stack of tropical linear layers by at most $\epsilon$ [71, 72].

# C  Proofs

*Proof of lemma 3.1.* If $\phi_\lambda$ is a valuation map, hence for all $a, b \in \mathbb{R}$, we have

1. $v_\lambda(0) = -\infty$;

2. $v_\lambda(ab) = v_\lambda(a) + v_\lambda(b) - \lambda$;

3. $v_\lambda(a+b) \leq \max\{v_\lambda(a), v_\lambda(b)\}$.

Property (i) is immediate from the definition. For $a, b > 0$, (ii) follows from $\log(ab) = \log a + \log b$:

$$v_\lambda(ab) = \log(ab) - \lambda = (\log a - \lambda) + (\log b - \lambda) + \lambda = v_\lambda(a) + v_\lambda(b) - \lambda.$$

If either factor is non-positive, both sides equal $-\infty$. For (iii), note that when $a, b > 0$ we have

$$\log(a+b) \leq \max\{\log a, \log b\} + \log 2,$$

so subtracting $\lambda$ preserves the inequality; if $a \leq 0$ or $b \leq 0$ the claim is trivial. Adding back the constant $\lambda$ to $v_\lambda$ eliminates the offset in (ii) while leaving (i)–(iii) unchanged, yielding the classical valuation $\widetilde{v}$. Collecting the $d$ coordinate-wise maps gives the vector-valued projection $\phi_\lambda : \mathbb{R}^d \to \Delta^{d-1}$, which is therefore a valuation map up to the projective (constant-shift) equivalence native to tropical geometry. $\square$

The main theorem here establishes that MHTA is an expressive tropically universal approximator of max-plus dynamic programming for combinatorial optimization such that every function that can be computed by a finite max–plus circuit admits a realization by a finite-depth MHTA stack. The proof proceeds in three stages. First we show that a single head can act as a tropical max *gate*. Second, we demonstrate that an $H$-head block can realize a tropical map by computing finitely many such maxima in parallel. Finally, we prove by structural induction that stacking a finite number of blocks suffices to emulate an arbitrary max–plus circuit. With the first lemma we want to show that a single head can realize a *weighted tropical* max *gate*.

**Lemma C.1** (Head–level Weighted $\oplus$ gate). *Let $J$ be a finite index set and let $\{x_j\}_{j \in J} \subset \mathbb{T}$ and $\{w_j\}_{j \in J} \subset \mathbb{T}$. There exists an attention head $h^* \in [H]$, a query–token index $i^* \in [N] \setminus \{t(j) \mid j \in J\}$, and distinct seq indices $t(j) \in [N]$ such that, after one forward pass, the context returned at $i^*$ is equal to*

$$c_{i^*}^{(h^*)} = \bigoplus_{j \in J}(x_j \odot w_j) = \max_{j \in J}\{x_j + w_j\}. \tag{3}$$

*Proof.* For a fix $h^*$ and $i^*$, for every $j \in J$, let's select a distinct token position $t(j)$. Then one can define the *value* vectors by $\mathbf{v}_{t(j)}^{(h^*)} := x_j \odot w_j$ and $\mathbf{v}_r^{(h^*)} := -\infty$ for all $r \notin \{t(j)\}$. To enforce (3) it suffices to make $s_{i^* \, t(j)}^{(h^*)} = 0$ for $j \in J$ and $s_{i^* \, r}^{(h^*)} = -\infty$ otherwise, because then $c_{i^*}^{(h^*)} = \bigoplus_{j \in J}(0 \odot (x_j \odot w_j)) = \max_{j \in J}(x_j + w_j)$.

One can write every query / key vector in block form $u = (u^{(1)}, u^{(2)}) \in \mathbb{T}^{d_k - 1} \times \mathbb{T}$. Fix arbitrary first blocks $u^{(1)}$ and arrange

$$\mathbf{q}_{i^*}^{(h^*)} = (0, \ldots, 0, 0), \qquad \mathbf{k}_{t(j)}^{(h^*)} = (0, \ldots, 0, 0), \quad j \in J,$$

so that $d_{\mathbb{H}}(\mathbf{q}_{i^*}^{(h^*)}, \mathbf{k}_{t(j)}^{(h^*)}) = 0$ and hence $s_{i^* \, t(j)}^{(h^*)} = 0$. For every *irrelevant* token $r \notin \{t(j)\}$ set

$$\mathbf{k}_r^{(h^*)} = (0, \ldots, 0, -\Gamma_r), \qquad \Gamma_r \gg 0,$$

so that the last coordinate differs from that of the query by $\Gamma_r$; consequently $d_{\mathbb{H}}(\mathbf{q}_{i^*}^{(h^*)}, \mathbf{k}_r^{(h^*)}) = \Gamma_r$ and $s_{i^* \, r}^{(h^*)} = -\Gamma_r$. Choosing $\Gamma_r$ large enough drives the score to $-\infty$ in the semiring, ensuring that irrelevant tokens do not influence the context. Equation (3) follows. $\square$

**Lemma C.2** (Tropical affine layer). *Let $A \in \mathbb{T}^{M \times N}$ and $b \in \mathbb{T}^M$. Embed $x = (x_1, \ldots, x_N) \in \mathbb{T}^N$ as the values of tokens $t(1), \ldots, t(N)$ and add one* bias *token $i_b$ whose value is fixed to $0$. There exists an MHTA layer with $H = M$ heads and $d_k = 2$ such that, for each $m \in [M]$,*

$$c_{i_m}^{(m)} = \bigoplus_{j=1}^{N}(A_{mj} \odot x_j) \oplus b_m,$$

*where $i_m$ is the query token of head $m$.*

*Proof.* For $m \in [M]$ and head $h = m$, we can apply Lemma C.1 with $J = \{1, \ldots, N\}$, input $x_j$ and weights $A_{mj}$ to obtain $\bigoplus_j(A_{mj} \odot x_j)$. Let the bias relevant to every head by assigning its key identical to the query, whence $s_{i_m \, i_b}^{(m)} = 0$ for all $m$. Then, we give it value $b_m$ *in head $m$ alone* via $\mathbf{W}_V^{(m)}$. The context becomes the maximum of $\bigoplus_j(A_{mj} \odot x_j)$ and $b_m$, completing the proof. $\square$

**Definition C.1** (Tropical circuit [15]). *A tropical circuit is a finite directed acyclic graph whose source nodes are labelled by variables $z_1, \ldots, z_n \in \mathbb{T}$ and whose internal nodes are labelled either by the operation tropical addition $(u, v) \mapsto u \oplus v = \max\{u, v\}$ or by the operation tropical multiplication $(u, v) \mapsto u \odot v = u + v$. The circuit computes a map $f : \mathbb{T}^n \to \mathbb{T}^m$ whose $m$ outputs are designated sinks. A circuit is* layered *if every edge points from layer $\ell$ to layer $\ell + 1$ for some topological layering $\{\mathcal{L}_0, \ldots, \mathcal{L}_L\}$. We write $\mathrm{depth}(\mathcal{C}) = L$ and $\mathrm{size}(\mathcal{C}) = |\mathcal{C}|$ for the number of internal gates.*

Because tropical multiplication distributes over tropical addition, every such circuit computes a *tropical polynomial*, namely a tropical sum $\oplus$ of finitely many monomials, each monomial being a tropical product $\odot$ (classical summation) of a subset of the indeterminates plus a constant. A *tropical polynomial* in variables $z = (z_1, \ldots, z_n)$ has an expression of the form

$$P(z) = \bigoplus_{k=1}^{K}\left( c_k \odot \bigodot_{j=1}^{n} z_j^{\odot e_{kj}} \right) = \max_{k \leq K}\left\{ c_k + \sum_{j=1}^{n} e_{kj} z_j \right\},$$

where $c_k \in \mathbb{T}$ and $e_{kj} \in \mathbb{N}$. Thus $P$ is *already* the maximum of finitely many affine forms in $z$. Lemma C.2 therefore applies directly.

**Theorem C.3** (Single–layer universality for tropical polynomials). *Let $P : \mathbb{T}^n \to \mathbb{T}^m$ be a vector-valued tropical polynomial map whose $m$ coordinates are $P_\ell(z) = \bigoplus_{k \leq K_\ell}(A_{\ell k} \odot z) \oplus b_{\ell k}$. There exists a single MHTA layer with $H = \sum_{\ell=1}^{m} K_\ell$ heads and $d_k \geq 2$ whose* tropical *output (the collection of all head contexts before the de-valuation $\psi = \exp$) equals $P(z)$.*

*Proof.* For each output coordinate $\ell$ one can allocate $K_\ell$ heads, one per affine term $A_{\ell k} \odot z \oplus b_{\ell k}$. Lemma C.2 shows that affine map in head $(\ell, k)$, depositing its value at a fresh query token $i_{\ell k}$. Because the score of an irrelevant head is $-\infty$, the contexts written to those tokens are ignored by all other heads. Finally, putting an aggregation head per output $\ell$ whose query token reads all tokens $i_{\ell k}$ with *score* 0 and returns their $\oplus$, namely $\max_k(A_{\ell k} \odot z \oplus b_{\ell k}) = P_\ell(z)$. No de-valuation is applied inside the tropical computation, so the result equals $P(z)$ in the max–plus semiring. $\square$

**Corollary C.3.1** (Depth–$L$ universality). *Let $F : \mathbb{T}^n \to \mathbb{T}^m$ be the output of a layered tropical circuit of depth $L$. Then, there exists an MHTA stack of $L$ successive layers which, on every $x \in (\mathbb{R}_{>0})^n$, produces*

$$\mathbf{C}^{(L)}(x) = F(\mathrm{val}(x)).$$

*Proof.* We can apply Theorem C.3 to each $P^{(i)}$ in succession, feeding the contexts of layer $i$ (still in tropical form) as the inputs to layer $i + 1$. Because no Euclidean de-valuation occurs after all MHTA layers, the tropical composition is preserved. $\square$

**Theorem C.4** (Simulation of max–plus Dynamic Programs). *Let $(S, E)$ be a finite directed acyclic graph with $|S| = N$ nodes and weighted edges $\{w_{uv}\}_{(u,v) \in E} \subset \mathbb{T}$. For $t \in \mathbb{N}$ define*

$$d_v(t+1) = \bigoplus_{u:\,(u,v) \in E}\left( w_{uv} \odot d_u(t) \right), \qquad d_v(0) = \delta_{v, v_0},$$

*where $v_0 \in S$ is the source node. For every finite horizon $T$ there exists a MHTA of depth $T$ and $N$ heads per layer such that the token values at layer $t$ equal the vector $\big(d_v(t)\big)_{v \in S}$ for all $t \leq T$.*

*Proof.* If we label the tokens by the vertices of $S$, at layer $t$ we store $d_v(t)$ in the value field of token $v$. To obtain $d_v(t+1)$ let head $h = v$ whose query token is $v$. Then, one can apply Lemma C.1 with index set $J = \{ u \mid (u, v) \in E \}$, input scalars $x_u = d_u(t)$ and weights $w_{uv}$, thereby producing $d_v(t+1)$ as context at token $v$. Since every head acts on em disjoint query tokens, all $v \in S$ are updated in parallel. Repeating for $T$ layers unrolls the dynamic program, hence layer $T$ realizes the horizon-$T$ value vector. $\square$

Let $(\Gamma = (V, E))$ be a directed graph with $(|V| = n)$ and tropical weighted adjacency matrix $(D \in \mathbb{T}^{n \times n})$. For $k \geq 0$ the tropical power $D^{\odot k}$ encodes path weights of length $k$. Fix a horizon $T \in \mathbb{N}$ and set the finite Kleene star

$$D^{*(T)}; := ; \bigoplus_{k=0}^{T-1} D^{\odot k} \in \mathbb{T}^{n \times n}.$$

For $v_0 \in \mathbb{T}^n$ the horizon-T value vector is $v_T = D^{*(T)} \otimes v_0$.

**Theorem C.5** (MHTA learns tropical transitive closure). *Given $\Gamma = (V, E)$ with $|V| = n$ and adjacency $D \in \mathbb{T}^{n \times n}$, for every $T \in \mathbb{N}$ there exists an MHTA stack of depth $T$ with $n$ heads per layer and head dimension $d_k$ such that the token values at layer $t$ equal $v_t \in \mathbb{T}^n$ and*

$$v_{t+1} ;=; D \otimes v_t \quad (t = 0, 1, \ldots, T - 1).$$

*In particular, the output at layer $T$ equals $v_T = D^{*(T)} \otimes v_0$.*

*Proof.* Indexing tokens by $(V)$ and store $(v_t(u))$ as the value on token $(u)$ at depth $(t)$, we fix $(v \in V)$ and in layer $(t)$, we assign head $(h = v)$ whose query is token $(v)$. Then, we apply Lemma B.1 to the index set $(J_v = u \in V : (u, v) \in E)$ with inputs $(x_u = v_t(u))$ and weights $(w_{uv} = D_{vu})$. The head returns at its query token

$$\bigoplus_{u \in J_v} \big(w_{uv} \odot x_u\big) ;=; \max_{u:(u,v)\in E} D_{vu} + v_t(u) ;=; (D \otimes v_t)(v) ;=; v_{t+1}(v).$$

All $(v \in V)$ update in parallel because heads use disjoint query tokens; thus $(n)$ heads suffice in each layer. Iterating over $t = 0, \ldots, T-1$ yields $(v_T)$, and by the path–semiring semantics we have $(v_T = D^{*(T)} \otimes v_0)$. $\square$

**Corollary C.5.1.** *In a tropical semiring and under the assumption that no negative cycle exist, shortest paths exist and are cycle-free, so their lengths are realized by paths of at most $(n - 1)$ arcs; consequently the infinite Kleene star truncates at $(n - 1)$ [73]. Thus, every shortest path uses at most $(n - 1)$ arcs, hence*

$$D^* ;=; I \oplus D \oplus \cdots \oplus D^{\odot(n-1)} ;=; D^{*(n)}.$$

*Therefore the MHTA stack of with depth $T \geq n - 1$ learns $(D^* \otimes v_0)$.*

In the embedding space of MHTA, the feasible domain is a polyhedron in parameter space on which shortest paths exist and the Kleene star is well-defined. The MHTA stack thus implements Bellman steps in the parameter space, thus the combinatorics of feasible solutions are absorbed in the polyhedral stratification. In particular, the map $(v_0, \theta) \mapsto D^{*(T)}(\theta) \otimes v_0$ is a tropical polynomial, and the Tropical Attention has a sufficient ability to learn such maps without recurrence. Hence, in principle, MHTA can learn the closure map itself rather than the step-by-step recurrence.

# D  Comparison between vanilla attention and Tropical Attention

In this section, we compare the algorithmic view between vanilla attention and Tropical Attention.

---

**Algorithm 1** Comparison between vanilla attention and Tropical Attention

---

**function** ATTENTION($\mathbf{X} : n \times d$)
  $\mathbf{Q}, \mathbf{K}, \mathbf{V} \leftarrow$ linear($\mathbf{X}$).chunk(3)
  $\widetilde{\mathbf{A}} \leftarrow$ einsum($id,\ jd \rightarrow ij, \mathbf{Q}, \mathbf{K}$)
  $\mathbf{A} \leftarrow$ softmax($\widetilde{\mathbf{A}}/\sqrt{d}, -1$)
  $\mathbf{O} \leftarrow$ einsum($ij,\ jd \rightarrow id, \mathbf{A}, \mathbf{V}$)
  **return** linear($\mathbf{O}$)
**end function**

**function** TROP_ATTENTION($\mathbf{X} : n \times d$)
  $\mathbf{Q}', \mathbf{K}', \mathbf{V}' \leftarrow \log(\text{ReLU}(\text{linear}(\mathbf{X})))$_chunk(3)
  $\lambda \leftarrow \text{Parameter}(\mathbf{N})$
  $\mathbf{Q} \leftarrow \mathbf{Q}' - \lambda, \ \mathbf{K} \leftarrow \mathbf{K}' - \lambda, \ \mathbf{V} \leftarrow \mathbf{V}' - \lambda$
  $\mathbf{Q}_{btd} = \max_j\big(\mathbf{Q}_{btj} + W^{(Q)}_{dj}\big)$
  $\mathbf{K}_{btd} = \max_j\big(\mathbf{K}_{btj} + W^{(K)}_{dj}\big)$
  $\mathbf{V}_{btd} = \max_j\big(\mathbf{V}_{btj} + W^{(V)}_{dj}\big)$
  $\forall i, j : \mathbf{D}_{bij} \leftarrow \max_d\big(\mathbf{Q}_{bid} - \mathbf{K}_{bjd}\big) - \min_d\big(\mathbf{Q}_{bid} - \mathbf{K}_{bjd}\big)$
  $\mathbf{S} \leftarrow -\mathbf{D}$
  $\forall i, d : \mathbf{C}_{bid} \leftarrow \max_j\big(\mathbf{S}_{bij} + \mathbf{V}_{bjd}\big)$
  $\mathbf{O} \leftarrow \exp(\mathbf{C})$
  **return** linear($\mathbf{O}$)
**end function**

---

# E  Dataset Details

Our evaluation suite covers eleven canonical problems[2]:

**Floyd–Warshall Dataset.**  Each example is a weighted directed graph with nonnegative edge weights; we compute the all-pairs shortest-path distances with the Floyd–Warshall algorithm and use the resulting distance matrix as the regression target. Inputs are the flattened (zero-filled for missing edges) weight matrix with positional indices.

---

[2]Code used to generate data can be found in our public repository: https://github.com/Baran-phys/Tropical-Attention/blob/main/dataloaders.py

**QuickSelect Dataset.** Each example is an unsorted list of integers together with an order statistic $k$; the label is a token-wise binary mask marking all positions that contain the $k$-th smallest value. Inputs provide per-element features (value and $k$ or a normalized rank proxy) in the original list order.

**3SUM–Decision Dataset.** Each example consists of a list of integers and a target $T$; the label is 1 if any three distinct elements sum to $T$, and 0 otherwise. Inputs present per-token pairs $[x_i, T]$ so the model can condition each element on the common target.

**Subset–Sum–Decision Dataset.** Each example is a list of integers and a target $T$; the label indicates whether some subset sums exactly to $T$. Inputs again use per-token pairs $[x_i, T]$, and labels are computed by an exact dynamic-programming subset-sum solver.

**Balanced Partition Dataset.** Given a multiset of integers, the goal is to split it into two subsets with minimal absolute difference of sums; the label is a token-wise $0/1$ membership vector for one optimal subset (chosen deterministically). Ground truth is produced via a standard DP over achievable partial sums with a traceback to a canonical solution.

**0–1 Knapsack Dataset.** Each instance provides item pairs $(v_i, w_i)$ and a capacity $C$; the label is a token-wise $0/1$ mask for an optimal item set that maximizes total value under the weight budget. We compute ground truth with an exact dynamic-programming solver and repeat $C$ on every token as an input feature.

**Fractional Knapsack Dataset.** Each instance also provides $(v_i, w_i)$ and a capacity $C$; the label is a per-item fraction in $[0, 1]$ from the optimal fractional solution. Ground truth is computed by the canonical greedy algorithm that takes items in decreasing value-to-weight ratio.

**Convex Hull Dataset.** Each example is a set of 2-D points; the label is 1 for points on the convex hull and 0 otherwise. Hull membership is determined with a standard monotone chain construction.

**Strongly Connected Components (SCC) Dataset.** Each example is a directed graph; the label is a flattened $n \times n$ matrix where entry $(i, j)$ is 1 iff nodes $i$ and $j$ belong to the same strongly connected component. The graphs are generated Erdős–Rényi. However, to make the dataset more challenging, we also added curvature to the graph by adding communities to the graph. We obtain SCC memberships using a graph library routine and expose adjacency plus positional indices as inputs.

**Bin Packing Dataset.** Each example supplies item sizes and a single bin capacity; the label is a token-wise $0/1$ indicator of whether an item starts a new bin under First-Fit Decreasing applied to the clean sizes. Inputs include each item's size, the shared capacity, and a normalized position index after sorting.

**Minimum Coin Change (0/1) Dataset.** Each instance contains a multiset of coin values and a target amount $T$; the label is a token-wise $0/1$ mask for one optimal solution using the fewest coins (or the all-zero mask if no solution exists). Ground truth is computed with an exact dynamic-programming solver with traceback.

# F   Training & Evaluation Protocol

This appendix complements the experimental setup outlined in Sec. 4. We focus on the conceptual pipeline. The low-level engineering choices (e.g. logging cadence, file formats) are documented in the public code repository[3]. The primary packages utilized in constructing our experiment is Pytorch [74], Pandas and Scipy [75], SciKitLearn [76], and Numpy [77]. The basic workflow is described below:

1. **Dataset generation.** For the selected combinatorial task we generate input and output pairs using the hyperparameters in Table 5

2. **Model instantiation.** A shallow Transformer encoder—configured with 1 layer, 2 attention heads and hidden width 64—is equipped with one of three attention mechanisms: *Vanilla*, *Tropical*, or *Adaptive*.

3. **Optimization.** We train for $N_{\text{epoch}}$ epochs using AdamW ($10^{-3}$, constant, no warm-up). We use one NVIDIA Tesla V100 GPU to train each model. Models trained with a sufficiently large batch size (500) training over 10M samples, took approximately 2.5 minutes to train. For more memory intensive graph models, our training time was approximately 45 minutes given small batch sizes of 16. The objective is chosen per-task:

   - BCE with logits – pooled binary tasks,

---

[3] https://github.com/Baran-phys/Tropical-Attention/

- token-wise BCE,
- mean-squared error – regression tasks.

$N_{epoch} = 100$

4. **Evaluation.** After training we reload the final checkpoint, generate a new test set, and compute (i) mean loss for regression tasks and (ii) $F_1$ for classification tasks on the generated test set. We evaluate our models on in-distribution data (data generated using the same hyperparameters as during training) and on out-of-distribution (OOD) data using the hyperparameters described in Table 5 using the OOD protocol described in Section 4. For Length OOD, all models were trained on sequence length of 8 and we evaluated them at sequence length of 64, with the exception of the graph problems (FloydWarshall and SCC), which were evaluated on sequence length of 16. For Perturbative Noise OOD, each input was perturbed with probability 0.5 with a random integer sampled from the task's perturbative noise range.

Table 5: Training hyperparameters and data ranges for each combinatorial task. Each task was trained with 10M samples, learning rate of 0.0001, input sequence length of 8, and no perturbations. The ranges in the table are used to draw random integer values for the given parameter within the data generation portion of the training.

| Dataset | Epochs | Target Range | Weight Range | Value Range | OOD Value Range | Perturbative Noise Range |
|---|---|---|---|---|---|---|
| SubsetSumDecision | 100 | (1,10) | N/A | (-5,5) | (-20,20) | (10,30) |
| Knapsack | 100 | (10,20) | (1,10) | (1,10) | (11,21) | (10,30) |
| FractionalKnapsack | 100 | (10,20) | (1,10) | (1,10) | (11,21) | (1,5) |
| MinCoinChange | 100 | (10,20) | N/A | (1,10) | (11,21) | (1,5) |
| Quickselect | 100 | N/A | N/A | (1,10) | (11,21) | (1,5) |
| BalancedPartition | 100 | N/A | N/A | (1,10) | (11,100) | (10,30) |
| BinPacking | 100 | (10,30) | N/A | (1,10) | (11,100) | (10,30) |
| ConvexHull | 100 | N/A | N/A | (0,10) | (11,21) | (1,5) |
| ThreeSumDecision | 100 | (-75,75) | N/A | (-20,20) | (-375,375) | (40,60) |
| FloydWarshall | 100 | N/A | N/A | (1,15) | (16,30) | (1,10) |
| SCC [4] | 100 | N/A | N/A | 0.001 | 0.1 | N/A |

---

[4]SCC uses a connectivity probability rather than an integer input value, hence the small decimal for Value Ranges and N/A for perturbatiove noise range. For perturbative noise range the connectivity switches with given perturbation probability.

