# OpenReview forum: "Tropical Attention: Neural Algorithmic Reasoning for Combinatorial Algorithms"
_NeurIPS.cc/2025/Conference — NeurIPS 2025 poster_

### Official Review · Reviewer_XE7o · 2025-06-06

**Clarity:** 2
**Significance:** 2
**Originality:** 2
**Rating:** 4
**Confidence:** 3

**Summary:**

Dynamic programming algorithms usually correspond to circuits over a particular semiring -- (max, +), (min, max), etc. Therefore, any neural model that has to execute DP algorithms would need to perform the sharp min/max operations. However, Vanilla transformers relying on scaled dot product attention cannot approximate those operations outside of their training support -- as the size increases, the attention scores "disperse" (almost all attention coefficients on a given row become equal). For this reason, the paper proposes modifying the standard attention mechanism to a tropical attention mechanism, operating on the (max,+) semiring.

The paper also defines the tropical semiring, its associated operations, how a valuation map, to convert from Euclidean to tropical space, should behave, what Tropical projective space is, and what a distance metric on that projective space is. The distance metric possesses projective invariance (adding a constant to each vector's components will not change the distance metric) and non-expansiveness (Any tropical linear map would result in a lower or equal distance metric). Those properties are particularly favourable when it comes to value generalisation.

The authors proceed to define multi-head tropical attention (MHTA), utilising the above tools (e.g. the distance metric). The particular valuation map and the inverse rely on logarithmization and exponentiation, the attention scores are calculated using the distance metric, and the aggregation is performed on the (max,+) semiring (Imagine standard MHA with $\sum$ replaced with max and multiplication replaced with +). The authors prove (Theorem 3.2 and Theorem A.3/4) that a stack of $T$ MHTA layers ($N$ heads/layer) can simulate any dynamic programming DAG circuit of $N$ nodes and depth $T$.

The newly proposed attention mechanism is compared against two baselines (vanilla transformer and adaptive temperature) on 11 algorithms and 3 different Out-of-Distribution (OOD) generalisation scenarios (length OOD/value OOD/adversarial attack). Quantitative results are often improved across the three OOD scenarios. Attention heatmaps suggest that Tropical attention is able to produce sharper attention coefficients even at larger scales.

**Questions:**

I have left several things in "Weaknesses", clearly detailing what is minor and what is major, with clear actionables. Please make sure you address them as well.

   * From Appendix C, I was able to infer that, e.g. for graph problems, the adjacency matrix was flattened, but also that for SCC, the **directed graph is symmetrized**. This, from the POV of the algorithm, doesn't make sense... elaborate. Also, for SCC, does the value generalisation mean we change the edge probability? The appendix also doesn't mention how graphs are generated (Erdos-Renyi/Barabasi-Albert/etc.?)

      (Here's one good reason to use GDM's benchmarks)
* When would standard attention be preferred over tropical? Apart from this study being conducted on synthetic data, are there any other limitations to your approach?
* How does tropical attention perform when producing a solution? Currently, e.g., shortest path distances are provided, but they don't provide the optimal path(s)... In the case of knapsacks, only the optimal value is provided, but not which items to take...

   (Yet another reason to use GDM's benchmarks is that they evaluate on solutions provided, not on regressing towards the value of the solution, which in NAR is seen as a simpler problem)

* Is the $N$ for the graph size in Theorem 3.2 (L255) the same $N$ as the one used in L257? If not, what's the minimum number of heads/layer?
* Shouldn't the architecture be a recurrent one? After all, the larger the input problem size, the larger the DP circuit becomes, and the deeper the architecture should be. One way to model variable depth is recurrence.
* Heatmaps of length 1024 are provided, but results only show 8 to 64 generalisation. Why were other sizes omitted?
* Vanilla vs adaptive attention heatmaps look almost identical (esp. in Figure 1) Why do we not notice any increase in sharpness, that the original authors observed?
* If adaptive attention changes the temperature based on **problem size** (N.B. not values), why is it included in Value OOD? In the case of Value OOD, how was the adaptive parameter calculated? Appendix D doesn't report if size is changed for value OOD.
* For adversarial OOD, do the authors check if the perturbation changes the algorithm's output?
* Why is the attention map (almost) identical irrespective of size in Figure 2, bottom row? Shouldn't the heatmap be different for different inputs?

**Ethical Concerns:**

["NO or VERY MINOR ethics concerns only"]

**Final Justification:**

The authors did a strong rebuttal, hence I give weak accept.

I keep it weak because as the authors say, there are conceptual similarities to prior work. As I have not explored the topic of tropical attention myself, I am happy to decrease my confidence.

**Limitations:**

yes, although I don't think those are the key limitations...

**Paper Formatting Concerns:**

no major, only minor (occasionally missing an interval between word and citation)

**Quality:**

2

**Strengths And Weaknesses:**

# Strengths
* Clear motivation for the paper
* Theoretically founded architectural change (not a result of random architectural search)
* Universal approximation guarantees

# Weaknesses
I'll tag some of the weaknesses as "minor weakness" -- if your paper contained **only** minor weaknesses, you would be around "borderline accept" with chances for "accept" if you are very convincing during the rebuttal, that, e.g., those are not bugs but features. If I don't say explicitly that something is minor, please consider it a major issue.

* **We have gone full circle** (minor) -- It is hard not to notice that head outputs (equation right after L221) are $max$-aggregated and they very much resemble how a graph neural network (GNN) works, particularly a hardcoded one like pDAB [1]. The message function is the value of the neighbouring node minus the Hilbert distance. (btw, the Hilbert distance is piecewise linear, thus, the whole message function is linear and can be reliably approximated via a ReLU-based MLP [2], if you choose to learn it via gradient descent). The GNN aggregation function is the $max$ operator.

   We started from a transformer, replaced the attention mechanism with a tropical one, and ended up with a GNN with a max aggregator, an architecture known to perform well in NAR settings. I don't mean to diminish any contribution of yours, but I believe that the MHTA layer is very related to a max-aggregated GNN layer and this relationship should be elaborated/explored (and maybe exploited too!).
* **Missing related work** (maybe a bit less, but still minor) -- Although not published in one of our top 3 conferences (NeurIPS/ICML/ICLR), the relationship between tropical algebra and GNNs has already been *theoretically* explored and accepted at a peer-reviewed venue([3]; I have left a hyperlink to the PDF). In fact, the paper authors also use logarithmization and exponentiation as pre- and post-processing steps. Their proofs focus on GINs, and the "GNN" here is slightly different, but I believe the two papers share a lot in common and should be discussed.
* **Mathematically overloaded** -- I believe that in an effort to look theoretically sound, the authors have overcooked it. Particularly, the terminology could be made a bit more lightweight, in order to make the paper more accessible to a general audience. E.g. did we have to define amoebas of algebraic varieties (the definition of which, mind you, is assumed) in order to make a point that the logarithm makes a perfect valuation map? Please consider what parts are necessary to convey your message, what knowledge is assumed (and not defined) and improve the write-up accordingly.
* **Unclear plots** -- It is not immediately clear to me what the rows in the attention maps are. Also, it is not clear to me what the authors mean by "columns are the 8 largest keys by $l_2$ norm". Please explain how the heatmaps are created in detail.
* **Poor benchmarking** -- my major concerns lie here. Fixing **all** those is necessary for "borderline accept":
   * To start with, the authors do not use any of the NAR canonical benchmarks (CLRS-30/CLRS-Text), nor do they give an explanation as to why. Both of those benchmarks can be used for data generation (when the algorithms are already implemented, e.g. quickselect or SCC) or expressing new algorithms (e.g. knapsack). Moreover, although the baselines provided there are written in JAX, any algorithm definition/data generation pipelines are framework-independent.

      The benchmarks allow us, reviewers, to more easily assess your contributions. For example, they define particular length OOD ranges, so one can measure the performance gains even for models/approaches not explicitly included in your baselines. They also include more complicated metrics (predecessor accuracy, etc.), so they can be seen as more challenging than the benchmark seen here. They even fix the test set so that everyone uses the same test dataset.

      I strongly encourage the authors to use either of the two benchmarks -- the different OOD regimes can be achieved by choosing the appropriate sampler parameter. Alternatively, I expect a **very good** justification for why benchmarks released by Google DeepMind have been abandoned in favour of custom ones, particularly clarifying what drawbacks of the GDM's benchmarks we are avoiding here. "Deep learning framework" is not accepted as an answer here.


    * The authors omit detailing their architecture hyperparameter choice (heads/dimensionality/number of layers) in the main text. They also conveniently omit to specify in the main text whether they use a recurrent MHTA layer or if the depth is fixed. I was surprised to see in Appendix D that:
       * The architecture is of fixed depth (`experiments.py` confirms it, `SimpleTransformerModel` accepts `num_layers` arguments and loops over that many layers; the class also uses flags to specify if it's using tropical vs classical attention)
       * The depth is 1, the number of heads is 2 (`jobs_to_do_train/evaluate` confirm it)... **across all architectures**.

       According to the authors' theory, "For every finite horizon $T \in \mathbb{N}$ there exists an MHTA network of *depth $T$*, using exactly $N$ heads [...] whose token values at layer $t$ equal the DP state vector" ($N$ is the number of vertices in the DP circuit). I understand that the theory says nothing about non-existence, but in order for existence to be guaranteed, we need many more layers, and we would also need number of heads and layers to vary with problem size (If I my knowledge is correct, as the problem size increases the DP DAG's number of vertices increases too).

        **I cannot vote for anything but rejection with such a discrepancy between theory and experimental setup.**

        I also encourage the authors to re-evaluate **all** baselines where at least the number of layers is set to some fraction of the largest sizes observed during test time. None of the architectures should work with 1 layer and 2 heads, esp. when tested in an OOD regime.

    * No standard deviations are provided -- please, include at least a couple of seeds (even 3 is better than 1) and provide aggregate statistics
    * No GNN baselines are considered. The most powerful NAR models are often in the form of GNNs. Given my initial weaknesses listed, I do not see a reason to omit them.

[1] Ong, E., Huszár, F., Lio, P., & Veličković, P. (2024). Parallelising Differentiable Algorithms Removes the Scalar Bottleneck: A Case Study. In ICML 2024 Workshop on Differentiable Almost Everything: Differentiable Relaxations, Algorithms, Operators, and Simulators.

[2] Xu, K., Zhang, M., Li, J., Du, S. S., Kawarabayashi, K. I., & Jegelka, S. (2021, January). How neural networks extrapolate: from feedforward to graph neural networks. In International Conference on Learning Representations (ICLR).

[3] Landolfi, F., Bacciu, D., & Numeroso, D. (2023). A tropical view of graph neural networks. In Proceedings of the 33rd European Symposium on Artificial Neural Networks, Computational Intelligence and Machine Learning [(ESANN 2023)](https://www.esann.org/sites/default/files/proceedings/2023/ES2023-27.pdf).

---

> ### Author Rebuttal · Authors · 2025-07-31
>
> >We have gone full circle...
>
> We agree that equation you mention shows that a single MHTA head computes $h_i=\max_j\{\,v_j-d_H(q_i,k_j)\}$, an operation that can be written as message passing on the complete digraph with a max aggregator, yet two acts make this layer richer than a classical max-GNN: (i) the *edge weight* $d_H(q_i,k_j)$ is learned projections of both endpoints, so the effective graph structure is input-dependent rather than geometry dependent (this is also the classical debate that Transformers are special case of Graph Attentions or the other way around); (ii) all calculations are carried out inside the $(\max,+)$ semiring, thereby preserving projective invariance and 1-Lipschitz continuity in Tropical Geometry, that disappear when one approximates the same score with a conventional ReLU MLP, whose exact replication would demand exponentially many linear regions. In particular, for any two points $q,k\in\Bbb R^{d+1}$ with $d_H = \textrm{diam}(q-k)$ is the difference of the maximum and minimum of the linear forms $z\mapsto z_i$, it is linear on every cone of the normal fan of the permutohedron, hence $d_H(z)=z_{i_{\max}}-z_{i_{\min}}$ is affine there. The permutohedron’s fan decomposes $\Bbb R^{d+1}\big/\langle{\bf 1}\rangle$ into $(d+1)!$ such cones, one for each total ordering of the $d+1$ coordinates, so $d_H$ has exactly that many linear regions. A feed-forward ReLU network of depth $L$ and width $w$ can carve out at most $O(w^L)$ regions (Montúfar et al., 2014); therefore matching the factorial growth of $d_H$ with any constant depth forces $w\ge\Omega\!\bigl((d+1)!^{1/L}\bigr)$, which is already $\Omega(2^{d})$ for $L\le d$ by Stirling’s approximation (see Jukna, Tropical Circuit Complexity). In contrast, a single MHTA head implements $d_H$ with no blow-up in width since $\max$ and $\min$ are native primitives of the $(\max,+)$ semiring: we compute the two extremal coordinates in parallel and subtract, preserving projective invariance and avoiding the exponential cost that an ordinary ReLU-MLP would incur. Ultimately, we believe that the graph-theoretic view here actually complements our analysis, but far from reducing MHTA to a known GNN. Our study provides a fundamental, algebraic-geometric explanation for why max aggregation is so effective for neural algorithmic reasoning. The full tropical parameterization enhances the attention layer with provable expressivity, geometric inductive bias, and practical gains that conventional attention kernels have not been shown to possess.
>
> >Mathematically overloaded...
>
> Our intention in introducing concepts like amoebas was to provide deeper geometric intuition for the valuation map and to motivate the interested reader to explore the rich connections between neural networks and algebraic geometry. We see this work as a bridge between these fields and aimed to provide pointers for a specialized audience. However, we take your point that some of this may be overly specialized for a general ML audience.
> >Unclear plots...
> Due to the lack of space, we kindly refer you to our response to RehNV.
>
> >To start with, the authors do not use...
>
> In fact we adopted their implementations, samplers and OOD length ranges for the overlapping tasks (Quickselect, Floyd-Warshall and SCC). However, our study ultimately centers on set-based attention and target not only length OOD, but value and adversarial noise OOD generalization setups. Therefore, we target a different and complementary slice of the algorithmic-reasoning landscape than the Google DeepMind benchmarks were designed for. CLRS’s objectives emphasise pointer supervision on graph-structured inputs (predecessor vectors, adjacency tensors, etc.), which is ideal for graph-message-passing models but forces an additional, decoding head that obscures the phenomena we wish to isolate, namely how replacing softmax by Tropical kernels alters the inductive bias inside attention itself. Moreover, eight of the eleven tasks we evaluate are NP-hard/complete optimisation problems (which were our initial targets) absent from CLRS but critical for testing whether a model can extrapolate combinatorial structure rather than execute a fixed polynomial-time routine. These tasks demand global, set-wise aggregation and expose the polyhedral geometry. Implementing them required custom setups, so for methodological consistency we retained a single, unified pipeline, inspired by the paper (2407.02793), instead of stitching two frameworks together. Furthermore, it is very important to emphasize that **our intention is not to beat a specific benchmark but to introduce a new theoretical methodology to the community that tries to resolve a fundamental drawback of softmax attention following tools from algebraic geometry**. Looking ahead and as a sequel paper to this study, we are working on a Tropical Graph model, that then it provides a head-to-head comparison to the SOTA NAR models. For the present paper, restricting to (set) sequence-level tasks, while inheriting CLRS code wherever applicable, allowed us to deliver a clean, controlled test of the theoretical contribution.
> >According to the authors' theory...
>
> Thank you for this insightful question, that exactly gets to the heart of our model's expressive power. The theorem 3.2 only gives an **upper bound**, not a lower bound. We only state that depth = T and N heads are *sufficient* to simulate a horizon-T DP circuit; it does not say they are necessary. In fact, we even stress that the construction is without any architectural restriction, i.e. it is merely a capacity guarantee, not a minimal-complexity claim. Consequently, seeing the model work with depth one and two heads is not a contradiction at all. A finite-horizon DP can be expressed as a sequence of max-plus (associative and idempotent) linear updates, $v_{t+1} = M \otimes v_t$. The entire $T$-step horizon can be collapsed by computing the tropical transitive closure~(Kleene star), $M^* = I \oplus M \oplus ... \oplus M^{(T-1)}$, allowing the final state to be found in one-shot via $v_T = M* \otimes v_0$ (1904.01082 incorporates this observation to analyze parametric shortest-paths). A single tropical-attention head computes $max_j(S_{ij} + v_j)$, thus it has a sufficient ability to learn a weight matrix that represents the transitive closure. Thus, the entire complexity of the multi-step process (as the solution space of Tropical polyhedron) can be encoded in the geometry of this shallow learned map (multiplexed in the feature dimensions of a single layer). Intuitively, **it is learning to identify the vertices and faces of the polyhedron that defines the combinatorial solution in the Tropical projective space**. Finally, because these operations exist in a geometry naturally described by the projective Hilbert metric, the learned piecewise-linear structure of the map $M^*$ is robust to certain changes in scale and length, providing a strong basis for OOD generalization. Hence, a shallow model has a theoretical justification for its power to approximate DP circuits and generalize beyond its training regime.
> >No GNN baselines are considered...
>
> In this study we deliberately restricted baselines to softmax and adaptive softmax attention because the scientific question we want to isolate is: **What happens when we replace the softmax kernel of self-attention by a Tropical kernel, keeping everything else unchanged?** To keep that comparison clean and interpretable, we restricted baselines to the two main soft-max variants (standard and it's update, adaptive). As reviewers 2 and 3 also observed, our contribution is primarily fundamental, introducing tropical attention as a novel attention kernel. In this study, we do not intend to break any records on standard benchmarks for SOTA GNNs in NAR. We are interested in using combinatorial algorithmic reasoning tasks to showcase the OOD power of the tropical Attention in comparison to the vanilla and adaptive attention. We agree that evaluating tropical attention inside GNN architectures is a natural next step, and we view this paper as a foundation the community can build on to explore those richer variants.
> >From Appendix C...
>
> You are correct; symmetrizing the graph for a directed task like SCC is incorrect. This was a remnant of a previous experimental setup that focused on simple connectivity in undirected graphs and was mistakenly carried over. We have now corrected this procedure afterwards. Indeed the value generalization means changing the inclusion an edge probability 0.01 during training and 0.1 during value OOD evaluation. The graphs were generated Erdős–Rényi as well. However, to make the dataset more challenging, we also added curvature to the graph by adding communities to the graph to make the algorithmic task more difficult for the networks. Above (response to R3), we are reporting the updated results from our experiments.
> >Why is the attention map...
>
> The consistency of the attention map for the Tropical model across different sequence lengths is indeed the intended result and is an evidence for the paper's central claims on OOD generalization. Preserving the underlying geometry inherent to these combinatorial problems, the stable attention pattern demonstrates that the model has successfully learned the the underlying algorithm, structurally independent of the input sequence length. The model is not merely ignoring the input, rather, it has internalized the structure of the algorithm itself. The attention patterns for vanilla and adaptive softmax degraded and dispersed as sequence length increases, showing their inability to generalize algorithmically, whereas the Tropical model's stability provides strong evidence of having learned the sharp, scale-invariant reasoning required for such tasks.
>
> **(We could not finish our answers to your questions due to the insufficient remaining charecters here!)**

---

> ### Author Response · Authors · 2025-08-01
> **Response to Reviewer XE7o (part II)**
>
> >When would standard attention be preferred over tropical?...
>
> Thank you for your question. Currently, in generative tasks (LM), we have not developed any autoregressive decoding strategy which we believe is a limitation. Another limitation is adapting TA for axial type reasoning (multidimensional reasoning such as Axial Attention) where the self-attention has to be performed along individual axes (we are currently working on this).
>
> >How does tropical attention perform when producing a solution?...
>
> We thank the reviewer for this insightful question. Within the current version of the paper, indeed with the classification tasks we are performing solution reconstruction, while as the reviewers noticed with the regression tasks, the models are performing value prediction. Between the submission and the current discussion, we tried to adopt solution reconstruction for all tasks, while following CLRS30 on the overlapping ones. As a result, in the camera ready version of the paper, we will report these new results. In the following, you can find the new results across all tasks.
>
> ### Out-of-distribution Micro-$\mathrm{F}_1$ score under Length OOD test (top) and MSE for regression tasks (bottom).
> | Algorithm         |    Vanilla   |   Adaptive   |     Tropical     |
> | :---------------- | :----------: | :----------: | :--------------: |
> | ConvexHull        | 42.75 ± 2.06 | 48.25 ± 0.96 | **97.00 ± 1.15** |
> | Knapsack          | 41.06 ± 1.76 | 39.18 ± 2.59 | **60.00 ± 2.09** |
> | Quickselect       |  4.66 ± 5.98 | 22.89 ± 2.49 | **77.06 ± 3.78** |
> | BinPacking        | 60.75 ± 2.49 | 64.25 ± 1.09 | **66.01 ± 1.55** |
> | SCC               | 51.30 ± 3.91 | 56.50 ± 2.22 | **89.25 ± 3.49** |
> | SubsetSum         | 21.13 ± 2.45 | 22.75 ± 5.25 | **87.50 ± 6.45** |
> | BalancedPartition | 80.55 ± 2.91 | 91.90 ± 5.52 | **96.73 ± 3.50** |
> | 3SUM              | 80.00 ± 0.82 | 79.75 ± 0.50 | **82.75 ± 1.59** |
> | MinCoinChange     |  9.25 ± 1.86 | 17.98 ± 2.29 | **42.52 ± 1.47** |
> | :----------------- | :----------: | :---------: | :-------------: |
> | Floyd-Warshall     | 12.81 ± 4.03 | 1.31 ± 0.36 | **0.81 ± 0.08** |
> | FractionalKnapsack |  0.88 ± 0.06 | 0.86 ± 0.08 | **0.66 ± 0.10** |
>
> ### Out-of-distribution Micro-$\mathrm{F}_1$ score under Value OOD test (top) and MSE for regression tasks (bottom).
> | Algorithm         |      Vanilla     |     Adaptive     |     Tropical     |
> | :---------------- | :--------------: | :--------------: | :--------------: |
> | ConvexHull        |   22.75 ± 3.59   |   23.77 ± 3.10   | **34.25 ± 1.71** |
> | Knapsack          |   38.87 ± 3.43   |   26.92 ± 1.33   | **49.67 ± 2.01** |
> | Quickselect       |   74.22 ± 2.30   | **74.30 ± 1.99** |   71.10 ± 3.11   |
> | BinPacking        |   67.26 ± 3.70   |   74.23 ± 1.51   | **78.54 ± 1.89** |
> | SCC               |   78.51 ± 3.08   | **81.38 ± 2.62** |   74.86 ± 5.01   |
> | SubsetSum         |   34.75 ± 6.60   |   28.50 ± 10.12  | **79.25 ± 5.38** |
> | BalancedPartition | **63.40 ± 4.29** |   56.57 ± 1.18   |   55.76 ± 5.63   |
> | 3SUM              |   26.00 ± 3.16   | **26.25 ± 3.50** |   22.00 ± 2.16   |
> | MinCoinChange     |   23.64 ± 4.07   |    2.20 ± 1.13   | **33.18 ± 5.64** |
> | :----------------- | :----------: | :----------: | :-------------: |
> | Floyd-Warshall     | 87.68 ± 5.65 | 56.30 ± 3.04 |   55.30 ± 4.36  |
> | FractionalKnapsack |  0.24 ± 0.12 |  0.17 ± 0.03 | **0.08 ± 0.03** |
> ### Out-of-distribution Micro-$\mathrm{F}_1$ score under Perturbative Noise test (top) and MSE for regression tasks (bottom).
> | Algorithm         |    Vanilla   |     Adaptive     |      Tropical     |
> | :---------------- | :----------: | :--------------: | :---------------: |
> | ConvexHull        | 90.75 ± 2.22 |   91.00 ± 2.16   |  **96.00 ± 2.16** |
> | Knapsack          | 67.85 ± 3.19 |   68.36 ± 3.51   |  **74.67 ± 3.13** |
> | Quickselect       | 33.87 ± 7.11 |   34.82 ± 4.79   |  **57.22 ± 5.01** |
> | BinPacking        | 55.38 ± 5.10 |   60.64 ± 3.92   |  **61.19 ± 4.33** |
> | SCC               | 70.00 ± 5.98 | **71.33 ± 1.96** |    69.86 ± 4.17   |
> | SubsetSum         |  3.75 ± 1.50 |     3 ± 1.63     | **72.75 ± 10.01** |
> | BalancedPartition | 51.06 ± 2.66 |   57.06 ± 1.08   |  **57.29 ± 1.33** |
> | 3SUM              | 47.50 ± 9.47 |   49.25 ± 9.22   |  **65.25 ± 3.59** |
> | MinCoinChange     | 22.12 ± 2.75 |   18.44 ± 4.49   |  **33.75 ± 4.89** |
> | :----------------- | :---------: | :---------: | :-------------: |
> | Floyd-Warshall     | 7.54 ± 3.63 | 5.29 ± 2.56 | **4.39 ± 1.62** |
> | FractionalKnapsack | 0.05 ± 0.02 | 0.03 ± 0.01 | **0.02 ± 0.01** |

---

> ### Author Response · Authors · 2025-08-01
> **Response to Reviewer XE7o (part III)**
>
> > Is the $N$ for the graph size in Theorem 3.2...
>
> Thank you for this comment. Yes, indeed each head is formulated tie to a vertex. We will clarify at first mention in the camera-ready to eliminate the ambiguity. On the minimum number of heads/layer, as we discussed above, the main theorem provides an upper bound, not a lower bound. Its purpose is a capacity guarantee, and it does not claim that N heads are information-theoretically necessary. Our small model implicitly exploit this.
>
> > Shouldn't the architecture be a recurrent one?...
>
> As we detailed in our earlier response regarding our architecture's theoretical sufficiency, a recurrent architecture is not strictly necessary here. In essence, due to the properties of Tropical geometry, instead of simulating every step of the DP recurrence, the model learns to model the entire multi-step computation into the tropical projective space. We kindly refer you to our more detailed explanation of this mechanism in our response to your earlier question.
>
> > Heatmaps of length 1024 are provided,...
>
> The heatmaps are meant purely as diagnostics and interpretability tools where we want to visualise how sharp the attention is when the sequence is stretched. Training on length 8 and report performance on length 64 is a standard practice in NAR to showcase length generalization. An 8x extrapolation from the training length is a significant challenge by itself and, as our results show, is more than sufficient to **expose the performance gap** between Tropical Attention and the baselines while keeping the comparison fair.
>
> >Vanilla vs adaptive attention heatmaps...
>
> This finding is actually consistent with the results from the original paper that introduced adaptive softmax. In (2410.01104) the authors reported that adaptive-temperature softmax does lower attention entropy OOD, but typically by only around $15\%$ and their own vanilla/adaptive heat-maps also look alike after certain lengths. And our results reaffirm the fact **while adaptive attention may offer a marginal quantitative improvement, it does not fundamentally solve the attention-fading problem.**
>
> >If adaptive attention changes the temperature based on problem size...
>
> This is an excellent question. In (2410.01104) the adaptive-temperature softmax does not take problem size as an explicit input. It rescales each head’s logits according to the Shannon entropy of the vanilla softmax distribution that those logits produce. Entropy, then is a function of the values of the logits, so the adaptation is inherently value-sensitive, even when sequence length stays fixed. Problem size only affects the temperature indirectly, to the extent that longer sequences tend to yield higher-entropy attention patterns. Therefore it is suitable to evaluate the same adaptive rule under value OOD conditions.
>
> >For adversarial OOD...
>
> Very good question. Yes, for all tasks, the adversarial protocol was designed to ensure the perturbations were meaningful and would alter the ground-truth output of the algorithm. We did this by carefully selecting the perturbation probability and the range of noise for each specific task. We will clarify this methodology in the appendix for the camera-ready version of the paper.
>
>
> If our comments resolve your concerns we would appreciate if you would consider raising your score!

---

> ### Comment · Reviewer_XE7o · 2025-08-01
> **Official reply**
>
> > [...] Hence, a shallow model has a theoretical justification for its power to approximate DP circuits and generalize beyond its training regime.
>
> A shallow model is of constant time complexity. Even if we take into account that each token is a "computational node", the time complexity is $$O(n^2*C)$$.
>
> Our algorithms are in P, but some of them, e.g. Floyd-Warshal are cubic. Thus, for large enough $n$, your network will behave as a quadratic function, whereas FWs algorithm, as a cubic. If for large $n$ your approximation is still bounded, then we have a quadratic APSP FPTAS. The best FPTAS I've found is around $O(n^{2.3sth})$ (and there are extra terms in the big-O; [https://arxiv.org/abs/1907.11078](https://arxiv.org/abs/1907.11078)) and they are quite certain it can't be improved further... What does this imply? That, in order not to "break our mathematical universe", Tropical attention will produce worse and worse approximations as size increases, to the extent they won't be useful.
>
> I do admit, that I did not grasp well the part after the Kleene star clojure, for which I am happy to decrease my confidence. (and be outruled by the AC), but I do firmly believe you're not benchmarking those networks correctly. In any case, even if Tropical attention doesn't need recurrence, it should be benchmarked against all possiblities, including recurrent attention models.
>
> >  In this study, we do not intend to break any records on standard benchmarks for SOTA GNNs in NAR. [...]
>
> In that case, maybe the focus should have been on approximating  *individual steps* of each algorithm, rather complete algorithms. I don't disagree that SOTA is not the most important part, but I'd like to know if Tropical attention would make any difference in a recurrent setting.

---

> > ### Author Response · Authors · 2025-08-03
> > **Response to Reviewer XE7o**
> >
> > Before addressing the specific points below, let us reiterate the *scope* of our contribution so that our discussion stays aligned with the paper’s goal.
> >
> > What this paper **does not** claim
> >
> > 1. *A state-of-the-art solver for every dynamic-programming task.*
> > 2. *SOTA Neural Algorithmic Reasoning (NAR) results.*
> > 3. *The final, optimal formulation of tropical attention.*
> > 4. *That a *single-layer* MHTA is always the best approximation for combinatorial problems.*
> > 5.  *That single-shot Tropical Attention is better than recurrent Tropical Attention.*
> >
> > What we **do** claim
> >
> > **Softmax self-attention has a structural weakness, its collapses toward uniformity as context grows, and replacing the softmax kernel with a kernel grounded in tropical algebraic geometry provides a principled alternative that improves OOD length, value and adversarial perturbation generalisation. To best of our knowledge, this level of exploring Tropical Algebraic Geometry (not just pre-post processing arithmetic) to define a new reasoning architecture has not been done before.**
> >
> > Our experiments therefore focus on an “apples-to-apples” comparison: vanilla softmax  ↔  adaptive-softmax  ↔  tropical.
> >
> > - Nevertheless, your concern is valid, but assumes a uniform, constant-depth network that gives a fixed-factor approximation for arbitrary graph size $n$. That is *not* what we want to say. We are not proposing a single, constant-depth network that provides a fixed-error approximation for an *arbitrary* graph size $n$. Our theoretical construction has training time to grow with $n$, compiles the transitive closure into model parameters, then evaluates a cached map in $O(n^{2}d)$ time. Once $n$ exceeds the range on which the network was capable, accuracy does fall. So we are not proposing a new quadratic-time FPTAS for APSP; **we are comparing inductive biases under the same training regime**.
> > You are absolutely right that one layer MHTA becomes a poor proxy for FW as graph size keeps growing, that is why for example, we kept FW as a prediction task and not a classification (solution reconstruction) one. While a shallow model is *sufficient* here to expose the difference in inductive biases between Tropical and Softmax attention, our framework is not architecturally fixed. For more complex reasoning tasks such as Long-Range Arena (LRA) benchmark, we scale the model accordingly and use a 4-6 layer MHTA. **The primary goal of our experiments was to conduct a fair comparison, keeping architectural factors like depth constant to isolate and evaluate the direct impact of the Tropical Attention mechanism against the baselines.**
> >
> > - Our objective is *not* to out-perform specialised GNNs or produce a breakthrough APSP algorithm. It is to show, through a controlled kernel swap, that the **tropical geometry and Hilbert metric enhance attention with sharper, scale-robust behaviour than softmax can deliver**.  The empirical evidence in both combinatorial algorithmic tasks and the standard  Long-Range Arena (LRA) benchmark supports that claim.
> >
> > > In that case, maybe the focus should have been on approximating...
> >
> > Studying end-to-end execution reveals the phenomenon we most care about, and that is **OOD scale-robustness of attention scores** in the most direct way. Measuring step-wise loss would require teacher-forcing or SSL self-supervised signals at each layer, which could help the model with the correct intermediate states and partially masks the very attenuation problem we are investigating. Once the core kernel is established, such per-step supervision is a natural next step where our goal is to push Tropical attention toward SOTA NAR results, but it would have blurred the central comparison in this paper.
> >
> > The clarifications above and the new experiments will be integrated into the camera-ready version. If our comments resolve your concerns we would appreciate if you would consider raising your score!

---

> > > ### Comment · Reviewer_XE7o · 2025-08-03
> > >
> > > 1. Agreed
> > > 2. Agreed
> > > 3. Agreed
> > > 4. Agreed
> > > 5. Agreed
> > >
> > > > To best of our knowledge, this level of exploring Tropical Algebraic Geometry (not just pre-post processing arithmetic) to define a new reasoning architecture has not been done before.
> > >
> > > Slightly unsure, because the kernel ends up being a maxplus semiring, which does not fall too far away from a max-aggregated GNN. **But for that, I am happy for my confidence to be taken less seriously.**
> > >
> > > > Our experiments therefore focus on an “apples-to-apples” comparison [...]
> > >
> > > I disagree here: Your approach is not an "apple", your approach is a 3-course Michelin-star meal (and, yes, I do like the idea in general). Tropical attention is capable of approximating the closure, whereas the other two models are not. Why not compare vs a model that is capable of approximating the closure, see how far you are off and discuss pros (e.g. you'd end up much faster) and cons (you may end up with poorer approximations). I think that such an experiment would have been necessary for acceptance.
> > >
> > > > Once $n$ exceeds the range on which the network was capable, accuracy does fall.
> > >
> > > The reason I became very confused is statements like this one above. If we are training a shallow network to simulate any closure up to size $n$, then OOD experiments are simply not the right ones, never mind that this is an NAR paper.
> > >
> > > In any case, that's just my POV on science... If AC+all other reviewers agree it's an accept, I am not going to safeguard NeurIPS.

---

> > > > ### Author Response · Authors · 2025-08-04
> > > > **Response to Reviewer XE7o**
> > > >
> > > > We appreciate that you like our idea and thank you for pushing us to include models that are capable of approximating the closure. We have now trained a 32-step Universal Transformer (UT) (1807.03819) with dynamic halting (ACT), as a recurrent attention model, under the vanilla softmax and adaptive-temperature softmax kernels and evaluated them on the OOD length generalization setup.
> > > >
> > > > **Table 1: Out-of-distribution $\text{Micro-F}_1$ score (top) and MSE for regression tasks (bottom)  under Length OOD test**
> > > > | Algorithm | UT w/ ACT | Adaptive UT w/ ACT | Tropical |
> > > > | :--- | :--- | :--- | :--- |
> > > > | **ConvexHull** | 43.37 | 53.83 | **97.00 ± 1.15** |
> > > > | **Knapsack** | 54.57 | 55.04 | **60.00 ± 2.09** |
> > > > | **Quickselect** | 37.05 | 40.44 | **77.06 ± 3.78** |
> > > > | **BinPacking** | 64.07 | 63.28 | **66.01 ± 1.55** |
> > > > | **SCC** | 74.68 | 70.81 | **89.25 ± 3.49** |
> > > > | **SubsetSum** | 41.43 | 42.05 | **87.50 ± 6.45** |
> > > > | **BalancedPartition** | 80.01 | 91.13 | **96.73 ± 3.50** |
> > > > | **3SUM** | 81.12 | 81.67 | **82.75 ± 1.59** |
> > > > | **MinCoinChange** | 17.33 | 23.67 | **42.52 ± 1.47** |
> > > > | --- | --- | --- | --- |
> > > > | **Floyd-Warshall** | 7.59 | 0.97 | **0.81 ± 0.08** |
> > > > | **FractionalKnapsack** | 0.83 | 0.85 | **0.66 ± 0.10** |
> > > >
> > > > **Table 2: Compute Performance**
> > > > | Model | Inference Time/ Sample (CPU) | Inference Time/ Sample (GPU) | Parameters |
> > > > | :--- | :--- | :--- | :--- |
> > > > | Vanilla UT w/ ACT | 6.285 ms | 0.027 ms | 50,242 |
> > > > | Adaptive UT w/ ACT | 7.898 ms | 0.018 ms | 50,242 |
> > > > | **Tropical** | **1.949 ms** | **0.003 ms** | **40,961** |
> > > >
> > > > The new results demonstrate two key findings:
> > > > 1.  **Superior Performance:** Even when compared against an iterative attention class, the Tropical Attention model still achieves better OOD performance across all algorithmic tasks.
> > > > 2.  **Greater Efficiency:** Our model achieves these results while being **3x-9x faster** at inference and using **~20% fewer parameters** than the UT baselines.
> > > >
> > > > > The reason I became very confused is statements like this ...
> > > >
> > > > Sorry for the confusion caused by our previous statement. Let us clarify the core message.
> > > > Our primary claim is **not** that we are perfectly simulating a closure for an arbitrary size *n* (That is our next step to challenge the NAR SOTA models where they indeed do degrade being test far enough OOD). The key contribution is that the **inductive bias from tropical algebraic geometry provides a representation that is fundamentally more robust to changes in scale**. All our OOD experiments are designed to show that this degradation is dramatically slower for our model, which is a direct result of its superior architectural foundation for algorithmic reasoning.
> > > >
> > > > > Slightly unsure, because the kernel ends up being a maxplus semiring...
> > > >
> > > > We agree there are conceptual similarities (as we have discussed in the rebuttal). However, we believe our contribution remains distinct due to the end-to-end framework derived from tropical algebraic geometry, including the **Tropical Hilbert projective metric** in computing attention scores and context, and the formal **theoretical link to Tropical circuits**, which is a novel approach for attention-based architectures.
> > > >
> > > > We hope these new experiments and clarifications fully address the reviewer's concerns. We are grateful for the feedback, which has allowed us to substantially strengthen our paper's evaluation. If our comments resolve your concerns we would appreciate if you would consider raising your score!

---

> > > > > ### Comment · Reviewer_XE7o · 2025-08-04
> > > > > **Consider it done**
> > > > >
> > > > > I have adjusted my scoring.

---

### Official Review · Reviewer_De71 · 2025-06-13

**Clarity:** 4
**Significance:** 2
**Originality:** 4
**Rating:** 5
**Confidence:** 3

**Summary:**

This paper introduces Tropical attention. A new formulation of attention based on strong prior results in Tropical geometry.
The authors introduce tropical attention formally in Section 3  in Definition 3.1, after extensively covering background materials in Section 2. To me there are three key points to this new method: 1) replacing matrix multiplication with max-plus matrix multiplication to obtain the Q, K and V matrices, 2) the Hilbert projective metric instead of $QK^T$ and softmax, 3) max-plus matrix multiplication of the V matrix and taking exponent of output.
The authors then provide theoretical guarantees on its expressivity and application to dynamic programming problems, which are well motivated in prior sections.
Finally the authors provide empirical experiments on dynamic programming focused combinatorial tasks. The baselines used in these experiments are sensible and very competitive for these sorts of tasks using softmax attention. Finally, the authors analyse some attention maps for the problems finding tropical-attention scales best OOD.

**Questions:**

1. Was all data unique and newly generated by the authors for this study or was prior work such as https://arxiv.org/abs/2406.04229 used? If not why was new data required, as this limits comparison to other works?
2. In Appendix D it is noted that the models are trained for many epochs. As we can generate synthetic data this is an infinite data regime, did the authors experiment with all unique data instead of epoching?
3. In figure 3 caption it is stated “Left to Right on length 16 to 1024” but the figure titles say 8 to 1024, which is it?
4. In the limitations it is states the computation and memory overhead of tropical attention is higher, exactly how much higher is it?
5. What would the loss of a tropical attention model be compared to a softmax attention model when trained on general webtext? I highly recommend this to the authors during the rebuttal period to maximise impact of this work. A positive result here could increase my score, a negative would leave my score unchanged. Please note, “could” not “will”, I will take into account all rebuttal material during the rebuttal.

**Ethical Concerns:**

["NO or VERY MINOR ethics concerns only"]

**Final Justification:**

# Why not higher
- I don't think the novelty of using tropical attention in a transformer has "groundbreaking impact" in one or more areas of AI

# Why not lower
- The paper is or interest to the Algorithmic Reasoning and GNN communities to see their ideas through the lens of tropical attention.
- A detailed analysis of the impact of different attention mechanisms is of value and novel.
- The paper is clear, even if a little mathematically overloaded.
- There is a good level of reproducible evaluation in the paper.
- No ethical concerns

I would like to note my confidence of a 3 related to some of the mathematical results in the paper, as highlighted in my rebuttal response, however I am more confident in my assessment of the empirical side of this work which I believe to be worthy of acceptance.

**Limitations:**

Yes.

**Quality:**

4

**Strengths And Weaknesses:**

# Strengths
- The authors throughly introduce the tropical semi-ring and results required to understand their paper. Overall, the paper is very clearly written and results well presented with extensive citations for all claims made that are not presented in this paper.
- The authors offer many theoretical results which back up their claims about tropical attention.
- Competitive and often better accuracy results on dynamic programming tasks.

# Weaknesses
- There isn't too much focus on related works that study reasoning (length/value generalisation) for softmax attention and the tricks deployed to overcome the short comings of softmax attention.

- The loss/perplexity of a model trained on general webtext with tropical attention compared to one trained with softmax attention is not provided.

## Formatting Issues
- Sometimes there is a space before \citep and sometimes there isn’t, please be consistent.
- In the summary of contributions there the bold comments are inconsistently followed by full stops.
- In the summary of contributions point 3, there is inconsistent spacing between commas.
- Axis labels on figures 1 and 2 are tiny.
- A, B and C are referred to in the captions of figures 1 and 2 but not on the figures.

---

> ### Author Rebuttal · Authors · 2025-07-31
>
> > There isn't too much focus on...
>
> Thank you for this constructive comment. In the camera-ready version, we thoroughly discuss the wider premises of this research field and comment on them.
>
> > The loss/perplexity of a model trained on general webtext with tropical attention compared to one trained with softmax attention is not provided.
>
> We thank the reviewer for this excellent suggestion. Given the short time we had in rebuttal for experimentation, we have conducted new experiments on the **Long Range Arena (LRA) ** benchmark, a standard for testing transformers on long-sequence tasks across text, image, and math domains. As shown in the table below, our Tropical Transformer achieves highly competitive, SOTA results, placing second overall in average accuracy across the benchmark's tasks. This strong performance shows that the benefits of Tropical Attention are not only to combinatorial problems and that it stands as a viable and powerful mechanism for general-purpose sequence modeling.
>
> | **Models**      | **ListOps** | **Text** | **Retrieval** | **Image** | **Pathfinder** | **Avg.** | **Complexity**        |
> |:----------------|:-----------:|:--------:|:-------------:|:---------:|:--------------:|:--------:|:----------------------|
> | Transformer     | 36.37 | 64.27 | 57.46 | 42.44 | 71.40 | 54.39 | $\mathcal{O}(n^2)$ |
> | Longformer      | 35.63 | 62.85 | 56.89 | 42.22 | 69.71 | 53.46 | $\mathcal{O}(n)$   |
> | Linformer       | 35.70 | 53.94 | 52.27 | 38.56 | 76.34 | 51.36 | $\mathcal{O}(n)$   |
> | Reformer        | 37.27 | 56.10 | 53.40 | 38.07 | 68.50 | 50.67 | $\mathcal{O}(n)$   |
> | Performer       | 18.01 | 65.40 | 53.82 | 42.77 | 77.50 | 51.41 | $\mathcal{O}(n)$   |
> | AdaptiveSoft.   | 47.15 | *75.52* | 79.56 | 51.58 | 80.94 | 66.95 | $\mathcal{O}(n^2)$ |
> | Elliptical      | 37.8  | 65.6  | 80.3  | 40.2  | 73.2  | 61.24 | $\mathcal{O}(n^2)$ |
> | Fourierformer   | 40.73 | 75.02 | *85.35* | 53.17 | 83.43 | 67.54 | $\mathcal{O}(n\log n)$ |
> | MEGA            | *63.14* | **90.43** | **91.25** | **90.44** | *96.01* | **86.25** | $\mathcal{O}(n\log n)$ |
> | **Tropical**    | **68.65** | 70.13 | 64.82 | *60.04* | **97.33** | *72.79* | $\mathcal{O}(n^2)$ |
> > Formatting Issues...
>
> Thank you for such a detailed suggestions. We have revised and refined our manuscript accordingly.
>
> > Was all data unique and newly generated ...
>
> In the beginning, we have started our study with various (**NP-Hard/NP-Complete**) combinatorial family of algorithmic tasks not present in CLRS30. We adapted three common tasks (Quickselect, Floyd-Warshall, and SCC) later in our experiments. For the three algorithms common to both our work and CLRS, we closely followed their implementation. So, all the datasets for this study were newly generated. This was necessary due to two fundamental incompatibilities with the CLRS-30 benchmark. First, there is a mismatch in data modality, our framework is designed for sequence and set-based inputs, while CLRS is exclusively graph-based. Second, our study focus on evaluating all three specific types of out-of-distribution generalizations, Length, Value, and Adversarial OOD, protocols which are not designed in CLRS setup. We will clarify this in the camera-ready version of the paper.
>
> > In Appendix D it is noted that ...
>
> We appreciate this very constructive comment. Following your suggestion, we conducted new experiments with much larger datasets, 10M samples. These new results, detailed in the table below, reaffirm our original conclusions. In addition, all new experiments were conducted across four random seeds, and several tasks were updated from a predictive (regression) to a constructive (classification) format to provide a more direct assessment of algorithmic learning. We believe these changes, suggested by our valuable feedback, significantly enhance our findings.
>
> ### 1. Out-of-distribution Micro-$\mathrm{F}_1$ score under Length OOD test (top) and MSE for regression tasks (bottom).
> | Algorithm         |    Vanilla   |   Adaptive   |     Tropical     |
> | :---------------- | :----------: | :----------: | :--------------: |
> | ConvexHull        | 42.75 ± 2.06 | 48.25 ± 0.96 | **97.00 ± 1.15** |
> | Knapsack          | 41.06 ± 1.76 | 39.18 ± 2.59 | **60.00 ± 2.09** |
> | Quickselect       |  4.66 ± 5.98 | 22.89 ± 2.49 | **77.06 ± 3.78** |
> | BinPacking        | 60.75 ± 2.49 | 64.25 ± 1.09 | **66.01 ± 1.55** |
> | SCC               | 51.30 ± 3.91 | 56.50 ± 2.22 | **89.25 ± 3.49** |
> | SubsetSum         | 21.13 ± 2.45 | 22.75 ± 5.25 | **87.50 ± 6.45** |
> | BalancedPartition | 80.55 ± 2.91 | 91.90 ± 5.52 | **96.73 ± 3.50** |
> | 3SUM              | 80.00 ± 0.82 | 79.75 ± 0.50 | **82.75 ± 1.59** |
> | MinCoinChange     |  9.25 ± 1.86 | 17.98 ± 2.29 | **42.52 ± 1.47** |
> | :----------------- | :----------: | :---------: | :-------------: |
> | Floyd-Warshall     | 12.81 ± 4.03 | 1.31 ± 0.36 | **0.81 ± 0.08** |
> | FractionalKnapsack |  0.88 ± 0.06 | 0.86 ± 0.08 | **0.66 ± 0.10** |
>
> ### 2. Out-of-distribution Micro-$\mathrm{F}_1$ score under Value OOD test (top) and MSE for regression tasks (bottom).
>
> | Algorithm         |      Vanilla     |     Adaptive     |     Tropical     |
> | :---------------- | :--------------: | :--------------: | :--------------: |
> | ConvexHull        |   22.75 ± 3.59   |   23.77 ± 3.10   | **34.25 ± 1.71** |
> | Knapsack          |   38.87 ± 3.43   |   26.92 ± 1.33   | **49.67 ± 2.01** |
> | Quickselect       |   74.22 ± 2.30   | **74.30 ± 1.99** |   71.10 ± 3.11   |
> | BinPacking        |   67.26 ± 3.70   |   74.23 ± 1.51   | **78.54 ± 1.89** |
> | SCC               |   78.51 ± 3.08   | **81.38 ± 2.62** |   74.86 ± 5.01   |
> | SubsetSum         |   34.75 ± 6.60   |   28.50 ± 10.12  | **79.25 ± 5.38** |
> | BalancedPartition | **63.40 ± 4.29** |   56.57 ± 1.18   |   55.76 ± 5.63   |
> | 3SUM              |   26.00 ± 3.16   | **26.25 ± 3.50** |   22.00 ± 2.16   |
> | MinCoinChange     |   23.64 ± 4.07   |    2.20 ± 1.13   | **33.18 ± 5.64** |
> | :----------------- | :----------: | :----------: | :-------------: |
> | Floyd-Warshall     | 87.68 ± 5.65 | 56.30 ± 3.04 |   55.30 ± 4.36  |
> | FractionalKnapsack |  0.24 ± 0.12 |  0.17 ± 0.03 | **0.08 ± 0.03** |
>
> ### 3. Out-of-distribution Micro-$\mathrm{F}_1$ score under Perturbative Noise test (top) and MSE for regression tasks (bottom).
>
> | Algorithm         |    Vanilla   |     Adaptive     |      Tropical     |
> | :---------------- | :----------: | :--------------: | :---------------: |
> | ConvexHull        | 90.75 ± 2.22 |   91.00 ± 2.16   |  **96.00 ± 2.16** |
> | Knapsack          | 67.85 ± 3.19 |   68.36 ± 3.51   |  **74.67 ± 3.13** |
> | Quickselect       | 33.87 ± 7.11 |   34.82 ± 4.79   |  **57.22 ± 5.01** |
> | BinPacking        | 55.38 ± 5.10 |   60.64 ± 3.92   |  **61.19 ± 4.33** |
> | SCC               | 70.00 ± 5.98 | **71.33 ± 1.96** |    69.86 ± 4.17   |
> | SubsetSum         |  3.75 ± 1.50 |     3 ± 1.63     | **72.75 ± 10.01** |
> | BalancedPartition | 51.06 ± 2.66 |   57.06 ± 1.08   |  **57.29 ± 1.33** |
> | 3SUM              | 47.50 ± 9.47 |   49.25 ± 9.22   |  **65.25 ± 3.59** |
> | MinCoinChange     | 22.12 ± 2.75 |   18.44 ± 4.49   |  **33.75 ± 4.89** |
> | :----------------- | :---------: | :---------: | :-------------: |
> | Floyd-Warshall     | 7.54 ± 3.63 | 5.29 ± 2.56 | **4.39 ± 1.62** |
> | FractionalKnapsack | 0.05 ± 0.02 | 0.03 ± 0.01 | **0.02 ± 0.01** |
>
> > In figure 3 caption it is stated ``Left to Right on length 16 to 1024'' ...
>
> We apologize for this confusion. The caption is correct. the evaluation was performed on out-of-distribution sequences of length 16 to 1024. The mention of length 8 refers to the training sequence length. We will revise the caption in the final version.
>
> > In the limitations it is states ...
>
> Our results shows that for shorter contexts, 64-128), our method is computationally on par with standard softmax attention, showing some negligible latency overhead. The significant increase in both computational and memory resources for longer sequences, $\mathcal{O}(1000)$, which is a consequence of the max/min implementation's complexity, that computes the tropical projective distance matrix. We believe that this engineering gap can be filled via the development of custom kernels or linearization tricks for tropical operations to achieving performance parity. Our main focus in this foundational paper was to establish a novel theoretical viability and unique properties of Tropical algebraic geometry for reasoning and combinatorial models, while introducing several NP-hard/complete algorithmic tasks (beyond CLRS30) to the community. To best of our knowledge, this is the first study that not just use Tropical arithmetic, but incorporate deeper Tropical algebraic geometric tools to offer a fresh perspective on Attention mechanism.
>
> > What would the loss of a tropical attention ...
>
> Thank you for this comment that made us motivated to push our results. As detailed in our response to your previous question, we run new experiments beyond the paper's original scope and evaluated our Tropical Transformer on the Long Range Arena (LRA) benchmark.
>
> If our comments resolve your concerns we would appreciate if you would consider raising your score!

---

> > ### Comment · Reviewer_De71 · 2025-08-01
> > **Rebuttal Response**
> >
> > Thank you for the rebuttal, comprehensively answering my few concerns.
> >
> > I have also read through the other reviewers points, noting many points from XE7o. I agree with XE7o, that the paper was mathematically overloaded for my own comprehension and gave my confidence accordingly in my initial review. I also agree on the point a tropical recurrent transformer would be interesting to see. Although the authors note it is not "strictly necessary here", the ability to learn a solution and actually learning a solution are two different things; and it is fun to learn what is in either or both of these categories. I don't believe the lack of a tropical recurrent baseline impacts the core of this paper in a material way though, more of an extension.
> >
> > I like the key rebuttal point about the motivation of the paper "What happens when we replace the softmax kernel of self-attention by a Tropical kernel, keeping everything else unchanged?" Transformers can be viewed through the lens of GNNs but with the current pace of LLM research being so fast, changing one thing at a time to make transformers more GNN like is overlooked. I view this paper as more of a stepping stone between the two areas of research. I think the omission of GNN baselines here is not a core issue as transformer baselines are analyzed exactly because of this point about a controlled change of one architectural hyperparameter at a time.
> >
> > I will maintain my score, with a confidence of a 3 as I believe there is a section of the NeruIPS community who would appreciate the novelty of this paper. However, I would like to note that my confidence is a 3, it is possible I did not understand some of the mathematical connections that need to be made to make a full assessment of this work, I would include the point "discrepancy between theory and experimental setup." that XE7o highlights in this.

---

> > > ### Author Response · Authors · 2025-08-03
> > > **Response to Reviewer De71**
> > >
> > > Thank you for recognizing the contribution our work can make to the community. Below is how we will refine the paper and address your remaining remarks in the camera-ready version.
> > >
> > > - We will relocate the more specialised notions from tropical geometry to the appendix and leave in the main text only the definitions strictly required.
> > >
> > > - Theorem 3.2 provides only an *upper* bound: it shows that a shallow network *can* simulate a horizon-$T$ DP, not that deeper or recurrent architectures are unnecessary. Empirically, a single-layer MHTA already learns an approximate transitive-closure map, collapsing multi-step DP updates into one max-plus matrix, an effect rooted in the linearising power of tropical algebraic geometry (Brugallé & Shaw, Tropical Enumerative Geometry, 2023). We will add a concise remark clarifying this theoretical-empirical link. So, **there is no discrepancy between theory and experiment.**
> > >
> > > - Broader motivation and “stepping-stone” perspective: As you suggested, we have now applied the tropical kernel to the **Long Range Arena benchmark**; its strong performance there reinforces the idea that **tropical attention also provides a practical bridge to problems beyond synthetic algorithmic tasks**.
> > >
> > > We appreciate your constructive engagement and respect your decision to keep your score while noting your confidence level. The clearer exposition, added empirical results, and explicit discussion of recurrent aggregators should make the final paper accessible to a wider NeurIPS audience without diluting its novelty.
> > >
> > > Please let us know if any further clarification would be helpful.

---

> ### Comment · Reviewer_XE7o · 2025-08-02
> **Reviewer response**
>
> Dear reviewer,
>
> Thanks for acknowledging my review. I want to state, that indeed, the NAR community would appreciate the paper's ideas. However, when it comes to theory alone, as stated in my review, Tropical spaces have been explored (Tropical attention, maybe not, but the pre- and post-processes steps were very similar) in related works, so it's really the turning theory into practice that really sets this paper apart from my POV.
>
> What I expected, however, is that the authors either test it in the standard recurrent setting or test it in a "emulate a single step" regime (the latter is also not unheard of and wouldn't have shocked me).
>
> Seeing that the 1-step approximation results are better for their OOD regimes (which, if I understood tables 1-3 correctly, are not 4x, but around 2x) and concluding that it would be better for any recurrent/other model or stating science contradictory statements$^1$ like "a shallow model has theoretical justification to approximate DP circuits and generalise beyond its training regime"; "a recurrent architecture is not necessary here" do not give me confidence in the paper.
>
> I don't have such in-depth mathematical knowledge as the authors, and for that I am happy to be outruled by other reviewers+AC and I do acknowledge I might have overestimated my confidence.
>
> But I do know basic computer science and according to it, this paper contradicts itself!
>
> ---------------------------------------------
> $^1$ To my best understanding of the theory of algorithms, and to my best understanding of Stefanie Jegelka's works, both approximation schemes and generalisation of neural networks share one common thing and that is that they state the generalisation error would be bounded. If this work were also to produce bounded approximations, learn to approximate the $O(n^2*2^n)$ DP algorithm for TSP and profit.

---

### Official Review · Reviewer_rLzR · 2025-07-03

**Clarity:** 1
**Significance:** 2
**Originality:** 3
**Rating:** 3
**Confidence:** 3

**Summary:**

The paper introduces a radically redesigned attention mechanism, replacing matrix-vector multiplication with a structure in which every sum becomes a max, and every multiply becomes a sum. The authors formally define multi-head tropical attention and prove that it can express any max-plus dynamic program. Empirical results are presented on combinatorial tasks such as Floyd–Warshall and bin packing, demonstrating superior performance compared to standard softmax attention on synthetic data. The paper also investigates out-of-distribution generalization with respect to sequence length, value ranges, and adversarial perturbations, consistently showing strong performance from tropical attention across these scenarios.

**Questions:**

What avenue do the authors see for tropical attention to become a viable attention alternative in practice?  Are there any issues with optimization with tropical attention?

**Ethical Concerns:**

["NO or VERY MINOR ethics concerns only"]

**Final Justification:**

The paper proposes a novel idea. It is however written with very difficult to understand mathematical jargon.  The additional experiments in the rebuttal is the reason for increasing the score.

**Limitations:**

Yes

**Quality:**

2

**Strengths And Weaknesses:**

The paper addresses a timely and important question: softmax attention was not originally designed for algorithmic reasoning, but rather for natural language processing. Exploring how to modify attention to better suit algorithmic tasks is a compelling direction. The proposed approach is a truly radical reinvention of attention, challenging the fundamental role of matrix-vector multiplication, which is an extremely bold and ambitious step. Given the magnitude of this proposal, focusing on synthetic experiments seems reasonable at this stage.

---

> ### Author Rebuttal · Authors · 2025-07-31
>
> > What avenue do the authors see for tropical attention to become a viable attention alternative in practice? Are there any issues with optimization with tropical attention?
>
> Thank you for the question. We see Tropical Attention as a strong alternative to the Softmax-family of attention kernels, particularly in problems with a combinatorial nature (searching or enumerating solutions or constructing combinatorial objects) and a length/value extrapolation requirement (reasoning tasks). Beyond the eleven problems evaluated in our work (and the recent LRA Transformer benchmark), We are aware of several teams investigating it's application in Phylogenetic inference in biology, Jet reconstruction in particle physics, logistics routing problems and Video-language models. Our own ongoing work, for instance, involves applying tropical Transformers to a logistic problem in the inventory of the fleet of U.S. Naval aircrafts. Regarding optimization, one issue that we also discussed in the limitations is the computational overhead of Tropical Attention due to the max/min implementation's complexity. We believe that this engineering challenge can be approached via the development of custom kernels or linearization tricks for tropical operations to achieving flash performance. The focus of this foundational paper was to introduce the Tropical Attention mechanism, establish its theoretical expressiveness, and demonstrate its empirical effectiveness on combinatorial (NP-Hard/NP-complete) tasks, thereby motivating future efforts in practical optimization.
>
> If our comments resolve your concerns we would appreciate if you would consider raising your score!

---

> > ### Comment · Reviewer_rLzR · 2025-08-08
> >
> > The paper is truly quite charming, and with the long range arena benchmark it is much easier to evaluate its performance on a canonical task.  That tropical attention is competitive at all is shocking.  My primary problem with the paper is more a matter of writing.  The authors clearly opted on the side of "fancy mathematical jargon" rather than grounding the presentation simply as possible.  This is hard to fix without major rewrites.  I will however adjust my score.

---

> > > ### Author Response · Authors · 2025-08-09
> > > **Response to Reviewer rLzR**
> > >
> > > Thank you for the generous assessment. We agree the exposition should prioritize clarity. For the camera-ready, we will definitely move specialized tropical-geometry material to the appendix and streamline notation. We appreciate the score adjustment and will make the presentation as accessible as the results.

---

> ### Comment · Reviewer_XE7o · 2025-08-08
> **I second that**
>
> The "bad" reviewer here! I just wanted to swing by to say that for the camera ready, the authors can and should adjust the rewrite.

---

### Official Review · Reviewer_ehNV · 2025-07-07

**Clarity:** 2
**Significance:** 3
**Originality:** 3
**Rating:** 4
**Confidence:** 4

**Summary:**

In this work, the authors propose a new attention mechanism to improve the OOD performance of transformers when they are trained to solve certain algorithmic tasks. Motivated by the observation that it is difficult for the standard softmax attention to execute length-independent (or value-independent) argmax/argmin operations that many dynamic programming algorithms rely on, the authors introduce a new attention mechanism based on the tropical semiring arithmetic. The tropical semiring arithmetic consists of binary maximum and binary plus operations, and hence an attention mechanism based on the tropical semiring arithmetic can naturally represent the recurrence structure in a dynamic programming algorithm for solving combinatorial problems. The authors carry out experiments on 11 combinatorial tasks and show that transformers equipped with the tropical attention mechanism achieve significantly better OOD performance.

**Questions:**

none

**Ethical Concerns:**

["NO or VERY MINOR ethics concerns only"]

**Final Justification:**

The authors have addressed my main concerns in my initial review. I increased my score accordingly.

**Limitations:**

yes

**Quality:**

2

**Strengths And Weaknesses:**

I think that the idea of using a different set of arithmetics in the computation of the attention matrix is definitely interesting. The motivation of this work is clear and intuitive.

However, the results of this paper are not surprising at all. Indeed, many of those combinatorial tasks considered in this paper can be solved by dynamic programming, and by construction, the Tropical Transformer architecture naturally represent some basic arithmetic operations (e.g. max-plus) in dynamic programming algorithms for these tasks. Beyond those synthetic combinatorial tasks, it is unclear whether or not the Tropical Transformer architecture will be useful for other more realistic tasks. The authors should really stress test the proposed architecture for more realistic tasks such as sequence modelling and natural language processing. Indeed, if the goal is to train a neural network to solve synthetic combinatorial tasks which can be solved by dynamic programming, why don't we just parameterize a recurrent network to mimic dynamic programming, and why is attention or the transformer architecture even relevant here?

Therefore, even though I think that this work has a clear motivation, the problem setting is very narrow (i.e. restricted to solving simple algorithmic tasks) and I am not sure if the findings in this work add any value to the community. The superior OOD performance might be simply due to a better algorithmic alignment between the Tropical Transformer architecture and the dynamic programming algorithms that solve those synthetic combinatorial tasks. In neural algorithmic reasoning, this benefit of aligning to specific algorithmic structures is pretty well-known already.

In addition, the presentation of the empirical results is not very clear. For example, Figure 1 is difficult to understand, and I am not sure how the authors reach to the conclusion that Figure 1 shows "the tropical attention maintains focus". Based on what I can see, the tropical attention pattern still keeps changing from the left to the right.

Finally, there are many typos in the paper. I recommend the authors spend serious effort to polish the writing, and add additional experiments on tasks beyond simple synthetic settings.

=================

The authors did a very good job during the rebuttal by providing additional experiments on more complex tasks. I have adjusted my score accordingly.

---

> ### Author Rebuttal · Authors · 2025-07-31
>
> > The results of this paper are not surprising at all....
>
> Our contribution extends beyond both DP tasks and "mere arithmetic." Several of our evaluated tasks, such as Quickselect, 3SUM-Decision, Bin-packing and Convex Hull, are not solved by dynamic programming, yet Tropical Attention excels on them as well. More importantly, our work introduces a novel algebrogeometric framework to look at the geometry of reasoning kernels from a fresh perspective. Also, Tropical Attention is not only exchanging arithmetic, every component of it is based on both Tropical Algebra/Geometry, and is beyond mere arithmetic. This geometric perspective is the foundational contribution of our paper.
>
> > Beyond those synthetic combinatorial tasks...
>
> While the tasks are synthetically generated, they are far from simple. They represent fundamental combinatorial problems, some of them NP-Hard/Complete, that are notoriously difficult for standard architectures and were chosen specifically to stress-test OOD reasoning capabilities. For instance, prior work (2409.07154) has shown that even a task like Quickselect is extremely challenging for state-of-the-art models (we discussed this in sec. 2.2). By succeeding on these difficult foundational tasks, we demonstrate a core generalization capability that current models lack. Moreover, these algorithms are the core mathematical pillars (hence the neural modeling bottleneck) of many real world combinatorial optimizations tasks, from particle physics (jet reconstruction) and biology (phylogenetics) to routing and logistics (dynamic bandwidth allocation in wireless networks, online electric vehicle charging in smart grids), and Finance (online trading problems in financial markets). Finally, to provide further evidence of its general utility, we also evaluated our model on the **Long Range Arena (LRA)** benchmark, a standard for testing transformers on long-sequence tasks across text, image, and math domains. As shown in the table below, our Tropical Transformer achieves highly competitive, SOTA results, placing second overall in average accuracy across the benchmark's tasks. This strong performance shows that the benefits of Tropical Attention are not only to combinatorial problems and that it stands as a viable and powerful mechanism for general-purpose sequence modeling.
>
> | **Models**      | **ListOps** | **Text** | **Retrieval** | **Image** | **Pathfinder** | **Avg.** | **Complexity**        |
> |:----------------|:-----------:|:--------:|:-------------:|:---------:|:--------------:|:--------:|:----------------------|
> | Transformer     | 36.37 | 64.27 | 57.46 | 42.44 | 71.40 | 54.39 | $\mathcal{O}(n^2)$ |
> | Longformer      | 35.63 | 62.85 | 56.89 | 42.22 | 69.71 | 53.46 | $\mathcal{O}(n)$   |
> | Linformer       | 35.70 | 53.94 | 52.27 | 38.56 | 76.34 | 51.36 | $\mathcal{O}(n)$   |
> | Reformer        | 37.27 | 56.10 | 53.40 | 38.07 | 68.50 | 50.67 | $\mathcal{O}(n)$   |
> | Performer       | 18.01 | 65.40 | 53.82 | 42.77 | 77.50 | 51.41 | $\mathcal{O}(n)$   |
> | AdaptiveSoft.   | 47.15 | *75.52* | 79.56 | 51.58 | 80.94 | 66.95 | $\mathcal{O}(n^2)$ |
> | Elliptical      | 37.8  | 65.6  | 80.3  | 40.2  | 73.2  | 61.24 | $\mathcal{O}(n^2)$ |
> | Fourierformer   | 40.73 | 75.02 | *85.35* | 53.17 | 83.43 | 67.54 | $\mathcal{O}(n\log n)$ |
> | MEGA            | *63.14* | **90.43** | **91.25** | **90.44** | *96.01* | **86.25** | $\mathcal{O}(n\log n)$ |
> | **Tropical**    | **68.65** | 70.13 | 64.82 | *60.04* | **97.33** | *72.79* | $\mathcal{O}(n^2)$ |
>
>
> > Indeed, if the goal is to train a neural...
>
> This is not the motivation and the goal of our work. We are trying to say that Softmax attention is flawed (2402.08164) and in order to go beyond the limits of it, one has to look at the problem from a different perspective. Our idea is that combinatorial algorithms - whether constructive, enumerative or search algorithms - have an important inductive bias resulting in a *sharp, polyhedral, hyperplane decision boundary*. Also, learning from Algebraic Geometry, where one linearizes the combinatorial computations using tropicalization and also by the fact that tropical circuits do have an expressive nature, we propose to map the reasoning core of Transformers (or any other attention based models), which is the softmax attention computation, to Tropical projective space. Although dynamic programming is a partial motivation, MHTA is motivated by the much broader class of combinatorial optimization problems. We have revised our manuscript to make sure that our message is conveyed properly and clearly.
>
> > In addition, the presentation of the empirical results is not very clear...
>
> For Figure 1, the key takeaway is that **attention is not uniformly dispersed in tropical attention**. We directly follow the implementation and the discussion of Figure 2 in this recent study, (2410.01104) , which demonstrated that softmax attention disperses to the uniform distribution as the input size increases. We draw samples (per seq. length), and  calculate the attention scores from the correct target item (the query) to the top 8 keys, which are ranked by their vector norms. In Quickselect task the goal is to find the k-th smallest elements. For such tasks, *maintaining focus* means the ability to allocate a high attention score to the correct items, creating sharp spikes in the heatmap. On contrary, *losing focus* means the attention becomes spread out and disperse, across elements, with no single item receiving a high score. Figure 1 shows that the vanilla and adaptive softmax models (top two rows) exhibit attention fading, as sequence length increases, indicating their attention has dispersed. In contrast, the Tropical attention (bottom row) consistently shows bright, distinct bands, proving it continues to allocate sharp attention even at larger length sequences. Moreover, each panel evaluates the model on a new batch of independently drawn inputs of increasing length. Since the position of the target k-th element is different in each batch, the pattern of attention naturally changes to reflect the new data. We have corrected many typos and will make the camera-ready version of the manuscript more clear upon revision based on the comments of the reviewers.
>
> If our comments resolve your concerns we would appreciate if you would consider raising your score!

---

> > ### Comment · Reviewer_ehNV · 2025-08-05
> >
> > I thank the authors for their response. I have carefully read all reviews and their responses. The authors did a good job in their rebuttal. The experiments on LRA benchmark added a lot of value to the work. The authors also properly addressed my initial misunderstandings. I will increase my score.

---

### Author Response · Authors · 2025-08-09
**Summary of Rebuttal and Discussion Period**

Dear AC, SAC, and Reviewers,

We thank the reviewers for the thoughtful reviews and the engaged discussion. We carefully addressed all points and substantially updated the manuscript (clarified theory, added new baselines and large-scale experiments, fixed dataset issues, improved figures/formatting).

## Main Contributions

* We introduce **Tropical Attention**, an attention kernel built in the $(\max,+)$ tropical semiring with a valuation to **tropical projective space** and reasoning core based on the **Hilbert projective metric** that learns a **tropical transitive closure**. This replaces softmax’s exponential normalization with a geometry that carries inductive biases that directly addresses attention *dispersion/fading* with scale.

- **Expressivity for algorithmic computation.** We prove that stacks of Multi-Head Tropical Attention (MHTA) can simulate max-plus DP circuits (upper-bound capacity), offering an *algebraic–geometric* rationale for sharp, length/value-robust behavior that softmax lacks.

- **Robust OOD generalization across three axes.** On 11combinatorial algorithmic tasks (most of which are NP-hard/complete), MHTA consistently outperforms softmax/adaptive-softmax under **Length-OOD, Value-OOD, and Adversarial-OOD**. Attention maps show MHTA **maintains focus** as context grows, unlike softmax variants.

- **Beyond algorithmic tasks.** New experiments (added in rebuttal) on the standard  **Long Range Arena (LRA)** benchmark, show MHTA is **highly competitive (second overall)**, towards general sequence modeling.

- **Recurrent baselines.** To address recurrence/closure concerns, we trained **Universal Transformers with ACT (32 steps)** under vanilla and adaptive softmax. **MHTA achieves much better OOD performance** while being **3–9× faster at inference** and using **\~20% fewer parameters**.

## Responses to Reviewers

- **Reviewer ehNV**
We clarified that our scope goes beyond DP and that these foundational algorithms underlie real domains (physics, biology, routing/logistics, finance). The new LRA results evidence broader utility. We clarified Figure-1 methodology and committed to language/typo cleanup.

- **Reviewer rLzR**
Reviewer called the idea of our paper a *truly radical reinvention of attention that tackles a timely and important question and shows superior performance*. On their question on applications and limitations, we reported that MHTA is near-parity at short contexts, with current overhead at longer lengths due to tropical distance computation. We also noted ongoing applications of MHTA (phylogenetics, jet reconstruction, routing, video-language; and logistics), and will further simplify exposition within the camera-ready version.

- **Reviewer De71**
We explained the story behind the choice of our dataset and why CLRS-30 do not align with our setup and OOD protocol, and added 10M-sample experiments. We clarified the Fig. 3 caption, discussed limitations, and acknowledged their great suggestion on testing over a standard sequence modeling benchmark, where we chose LRA as a cross-domain test.

- **Reviewer XE7o**
We had a great and productive discussion with them that raised the quality of our work. We addressed the max-GNN relation to our work. We will add missing GNN related work discussion and trimmed specialized background to the appendix. Most importantly, we added UT-ACT baselines where MHTA has a better OOD performance while being faster/smaller. We also converted several regression tasks to constructive (solution-reconstruction) classification tasks.

## Final Remarks:

Our study introduces Tropical Attention with a **controlled, theory-first experiment**, softmax → tropical in Transformers, yielding **consistent OOD gains** on NP-hard/complete algorithmic tasks, **competitive LRA** performance across domains, and recurrent attention (Universal Transformer) comparisons where MHTA leads while being more efficient. It **bridges transformers and GNN-style reasoning** through a rigorous tropical-geometric lens that explains why sharper, scale-robust attention emerges. **To best of our knowledge, this level of exploring Tropical Algebraic Geometry (not just pre-post processing arithmetic) to define a new reasoning architecture has not been done before.**

For the camera-ready, we will (i) streamline exposition (including the new discussion on closure), (ii) include LRA and UT-ACT tables and 10M sample results, (iii) document corrected datasets/protocols, (iv) present clearer figure captions and elaborations, and (v) expand related work.

We respectfully request that the area chairs and senior area chairs consider our comprehensive responses and the positive evaluation from all the reviewers when making their decision. We are confident that our work will be of interest and value to the NeurIPS community.

Thank you for your time and consideration. We remain available to provide any additional information or clarification as needed.

Best regards,
The Authors

---

### Decision · Program_Chairs · 2025-09-17

**Decision:**

Accept (poster)

**Comment:**

This paper proposes a simple and effective idea: by changing the semiring in attention mechanisms to the tropical semiring (rather than the conventional softmax-dot attention), attention-based neural architectures become better suited to represent, process and solve combinatorial optimization problems. The generalization performance of the approach is demonstrated on 11 combinatorial optimization tasks.

The paper's approach is simple, interesting and promising, and the experimental results are compelling. However, the experiments were perceived as somewhat toy by the reviewers. Also, the reviewers and the AC perceived the paper's presentation is somewhat "overloaded": The current exposition relies on a level of abstraction and formality that seems to go beyond what is necessary for the results. This risks obscuring the main ideas and may make the paper harder to follow than it needs to be. I recommend presenting the arguments in a more direct and concrete manner, with simpler notation and fewer technical detours, so that the central contributions are more transparent to the reader.